# TEDL: A Text Encryption Method Based on Deep Learning

**Peng Wang** [1] and **Xiang Li** [2],*

1 School of Computer Science and Engineering, Southeast University, Nanjing 211189, China; pwang@seu.edu.cn
2 Institute for Interdisciplinary Information Sciences, Tsinghua University, Beijing 100084, China
* Correspondence: lixiang20@mails.tsinghua.edu.cn

**Abstract:** Recent years have seen an increasing emphasis on information security, and various encryption methods have been proposed. However, for symmetric encryption methods, the well-known encryption techniques still rely on the key space to guarantee security and suffer from frequent key updating. Aiming to solve those problems, this paper proposes a novel symmetry-key method for text encryption based on deep learning called TEDL, where the secret key includes hyperparameters in the deep learning model and the core step of encryption is transforming input data into weights trained under hyperparameters. Firstly, both communication parties establish a word vector table by training a deep learning model according to specified hyperparameters. Then, a self-update codebook is constructed on the word vector table with the SHA-256 function and other tricks. When communication starts, encryption and decryption are equivalent to indexing and inverted indexing on the codebook, respectively, thus achieving the transformation between plaintext and ciphertext. Results of experiments and relevant analyses show that TEDL performs well for security, efficiency, generality, and has a lower demand for the frequency of key redistribution. Especially, as a supplement to current encryption methods, the time-consuming process of constructing a codebook increases the difficulty of brute-force attacks, meanwhile, it does not degrade the efficiency of communications.

**Keywords:** text encryption; deep learning; hyperparameter

## 1. Introduction

Today, more and more important data are transmitted in text format, whose security is guaranteed by various encryption methods. They include classic encryption algorithms (e.g., 3DES, AES, RSA) that have been widely used, as well as some innovative encryption algorithms (e.g., DNA algorithm [1,2], chaotic map algorithm [3]). Especially, AES, a representative of symmetric encryption, is rather popular and accepted as data encryption standard [4], due to its high speed and low space performance. Besides, another symmetric-key algorithm called one-time pad (OTP) [5,6] proves to be unbreakable. However, some defects still exist. Firsly, the strength of most symmetric-key algorithms relies on the key size [7,8]. It means that the security degrades proportionally as the key space gets smaller. One solution is to increase the complexity of the encryption algorithm. Then for the same size of key space, attackers need more time to crack. However, it sacrifices efficiency, namely, both communication parties need to spend more time on encryption and decryption. It is a challenge to achieve a balance between security and efficiency. Moreover, for OTP, when a large amount of information needs to be transmitted, it suffers from the difficulty in key updating. The problem of secure key distribution makes it impractical for most applications [9]. Therefore, cryptologists are constantly designing more practical encryption methods to get close to OTP. The stream cipher is one of the alternatives, while it is vulnerable if used incorrectly [10].

Deep learning [11] has become a hot field in artificial intelligence. By training, the learning model can automatically learn the mapping from massive data to the labels. This

process is controlled by some hyperparameters and generates a large number of unexplained parameters. Sometimes these parameters act as abstract representations of input data, although they do not seem to have any clear relations. When those hyperparameters are unknown or changed, or when the labels are altered, the exact parameters cannot be obtained. Therefore, a deep learning model embodies the nature of encryption. In other words, replacing meaningful data with corresponding parameters can be regarded as an encryption process [12]. A typical case is word embedding based on deep learning model [13,14], the cornerstone of Natural Language Processing (NLP). The most classic one is the *Word2vec* [15,16], which is improved by *Glove* [17], *fastText* [18], and so on. These models map words into distributed representations, which consist of parameters. Once we change the hyperparameters or corpus, the word representations (parameters) will change. In addition, deep learning training usually takes a long time, and the adjustment of each hyperparameter means a lot of time lapse. To some extent, this feature is useful to enhance security.

In this paper, we introduce the above characteristics of deep learning into text encryption and propose a novel symmetric encryption method for text encryption named TEDL. It adopts a public corpus as the original corpus, two copies of which are owned by both communication parties. They modify the copy at hand to obtain a synthetic corpus under the guidance of the key, respectively. The synthetic corpus owned by both should be confidential and consistent. After that, same word embedding models are used to train on the synthetic corpus under the hyperparameters specified by the key and construct word vector tables. Combined with the SHA-256 function [19], they are further processed to obtain time-varying codebooks, which is definitely consistent as well. The sender replaces the plaintext with a ciphertext based on the codebook and transmits it to the receiver. The receiver decrypts the ciphertext according to the codebook in turn.

The contributions of our paper are as follows:

- To the best of our knowledge, although there exists some work with respect to the combination of deep learning and information security [20–22], TEDL is among the first to utilize the uninterpretability and time-consuming training features in deep learning to realize encryption. Moreover, it is the first time that the word embedding based on deep learning model is used for encryption.
- TEDL has time-varying and self-updating characteristics, which are greatly beneficial for reducing the frequency of key redistribution. The time-varying refers to the variation of codebook as information is transmitted. Consequently, for the identical word, its representation varies every time. To some extent, it is close to the one-time pad. Besides, the concept of self-updating means that both sender and receiver can reconstruct codebook by revising synthetic corpus, without changing the key.
- TEDL is sensitive to the change of corpus. We prove that the process of skip-gram hierarchical softmax (SGHS) is equal to implicit matrix decomposition, beneficial to the better understanding of the word embedding process. Moreover, it means that minor changes in the corpus can cause a wide range of adjustments in training results, just as changes in a small number of elements in a matrix lead to a wide variation in matrix decomposition results. It directly supports the feasibility of our method.
- TEDL has a two-stage structure: codebook construction stage and communication stage. The former needs a long time, and the brute-force crack is performed at this stage. The latter is always time-saving, for it only involves a search operation. Furthermore, communication is mainly carried out at the second stage. In this way, both parties are able to achieve relatively high-speed communication while attackers still need more time to crack.
- TEDL performs well for security and high efficiency concluded by experiments and relevant analyses, which involve the recoverability, time cost for a brute-force attack, frequency analysis, correlation analysis, sensitivity analysis, efficiency, and generality.

The rest of the paper is organized as follows. In Sections 3 and 4, we outline our method and give some preliminaries, respectively. In Section 5, we illustrate the key design.

Section 6 details the encryption/decryption process. Section 7 introduces the self-updating mechanism in TEDL. We prove the feasibility of TEDL in Section 8 and the security analysis is shown in Section 9 with experiments in Section 10. Section 11 reveals the limitations of TEDL. Section 2 discusses the related work. Finally, Section 12 draws conclusions and further work.

## 2. Related Work

With the development of deep learning and increasing attention to information security, the application of deep learning in the field of information security has developed and extended. Perhaps the most common intersection between them has proven to be data privacy. After a series of transformations, the deep learning model can be trained and used for operations (e.g., prediction, classification) on encrypted data, in order to address the issue of access to sensitive raw data [23–25].

Although it is the first time to apply deep learning model, especially word embedding directly to encryption, prior to this, word embedding has been combined with other information security technologies, which mainly utilize the similarity between word vectors. For example, in [26], word vector is used on encrypted cloud data to achieve lightweight efficient multi-keyword ranked search by finding the most similar query vector and document vector. Besides, in [27], the original keyword is substituted by a similar keyword in case the text retrieval fails. This replacement is achieved by finding word vectors with high similarity. Instead, our method acts in a diametrically opposite way, where achieving low similarity between the representations of the same word is expected.

Although neural network-based generative sequence models unconsciously memorize secret information, as described in [28,29], that is, given models and data, there is a way to determine whether the data is used as part of the training data set, resulting in the disclosure of secret data. It is not the case for our method, where any word possibly used is definitely contained in training corpus but it is hard to determine whether the word is the corresponding plaintext. In contrast, we just take advantage of the characteristic, that is, the subtle changes in the corpus can be "memorized", assisting encryption.

In addition to [30–33] mentioned in Section 8, many papers also aim to issue the interpretability of model, such as [34–36]. Accordingly, some evaluation methods for interpretation have also been proposed [37]. However, these are not enough. Research on interpretability is important for both improving and cracking TEDL. In turn, the exploration of TEDL can promote the development of model interpretability.

## 3. TEDL Overview

It seems that security and efficiency are usually contradictory. Increased encryption algorithm complexity probably means strengthened security and reduced efficiency, which motivates people to search for the best trade-off. In this paper, our TEDL method provides a novel way to deal with this problem. As Figure 1 shows, TEDL contains two stages: (1) communication preparation and (2) communication process.

At the first stage, both parties in the communication get copies of the public corpus and modify them under the instruction of the key, completing the construction of confidential synthetic corpora, respectively. Furthermore, the synthetic corpora mastered by both parties are expected to be consistent. Afterward, the hyperparameters in the key instruct the training on the synthetic corpora. Hence word vector tables are established, followed by a further process on them with the SHA-256 function to obtain codebooks. To date, the first stage called communication preparation ends.

At the second stage, when a word requires transmitting, the sender refers to the codebook at hand and uses the plaintext as an index unit to obtain the corresponding ciphertext. Furthermore, then the ciphertext is sent to the receiver. In turn, the receiver decrypts the ciphertext based on the mapping in the codebook, which is equivalently an inverted indexing operation. After completing the transmission of a word, both ends adjust

the codebook in a certain way. Therefore, when the next word needs to be transmitted, it is encrypted based on the new codebook.

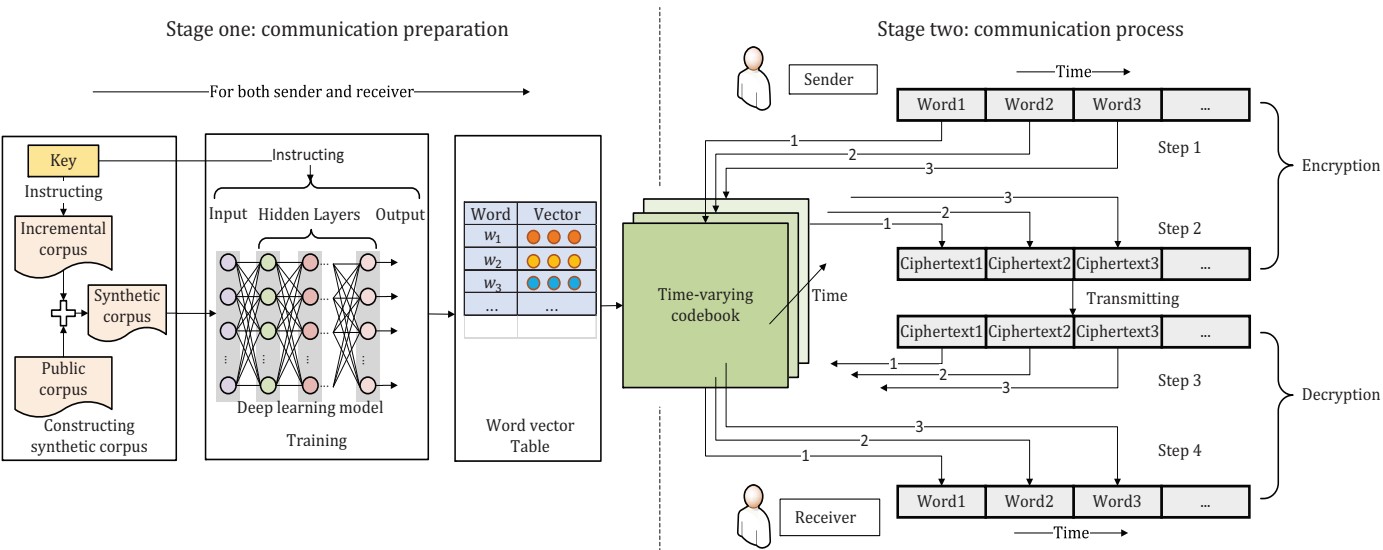

**Figure 1.** TEDL overview: a two-stage encryption method.

## 4. Preliminaries

We first give the symbols and fundamental definitions used throughout the paper as Table 1 lists. Here, we take an example illustrated in Figure 2 to explain some definitions. Given a corpus (Public corpus and Original corpus) such as selections from Shakespeare, we can generate word vectors with a word embedding model based on deep learning. As we add some additional text to the corpus, word vectors will change. Provided an ISBN number, 9780679405825 (Initial address), we can find the book *JANE EYRE*, from which we can select some content, denoted as $v_1^0$ (Initial incremental corpus unit), according to page number or chapter number. Then we can enlarge the initially selected content by including their adjacent pages or chapters, denoted as $v_j^i$ (Incremental corpus unit). Finally, all of them serve as the new corpus (Incremental corpus) to be added to the original corpus.

**Definition 1** (Public corpus). *A corpus can be obtained by anyone and should be chosen according to the language of the plaintext.*

It could be the Bible, Wikipedia (https://corpus.byu.edu/wiki/), iWeb (https://corpus.byu.edu/iweb/) and so on.

**Definition 2** (Original corpus). *Define $C_\alpha$ as either a public corpus or an expired synthetic corpus, which is a synthetic corpus generated under the guidance of the last key.*

**Definition 3** (Synthetic corpus). *Define $C_\gamma$ as a corpus obtained by revising the original corpus. Make sure the synthetic corpus contains words in the plaintext, otherwise, their corresponding ciphertext is not available.*

**Definition 4** (Initial address). *It gives the location of textual information and is part of the key.*

There exists various addresses, such as arXiv ID, uniform resource locator (URL), digital object identifier (DOI), International Standard Book Number (ISBN) and so on.

**Definition 5** (Initial incremental corpus unit). *Define $v_j^0$ as the text obtained from the initial address.*

**Definition 6** (Incremental corpus unit). *Define $v_j^i$ as the text that has a relationship (e.g.,citation, context) to the initial incremental corpus unit.*

**Definition 7** (Initial incremental corpus graph). *Define $G_\iota$ as an abstract structure inside the initial incremental corpus. It is a directed graph.*

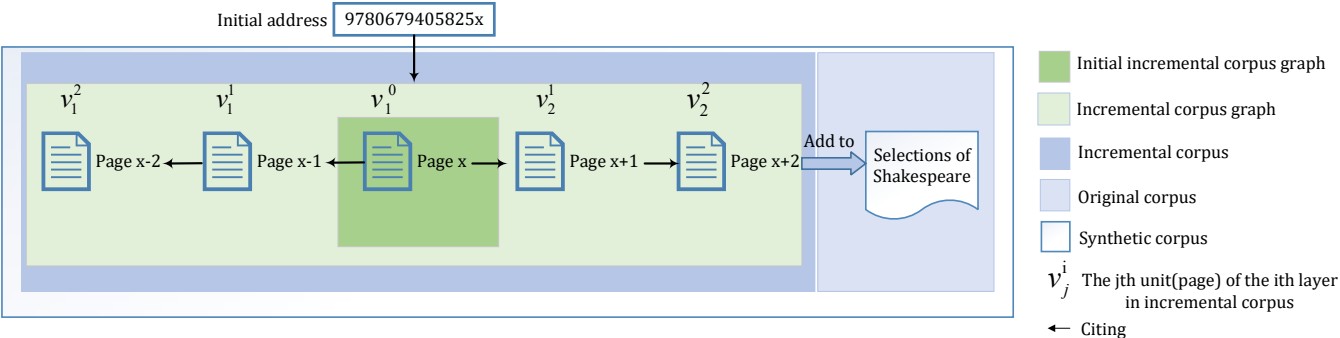

**Figure 2.** An example of using ISBN address to construct the synthetic corpus.

In the example, $V_\iota = v_1^0$ and $E_\iota = \varnothing$.

**Definition 8** (Incremental corpus graph). *Define $G_\beta$ as a directed graph that represents the structure inside the incremental corpus.*

**Definition 9** (Distance between node $u$ and $v$). *Define $d(u,v)$ as the length of the shortest directional path from vertex $u$ to vertex $v$.*

The distance between adjacent vertices (e.g., $v_1^0$ and one of its references $v_1^1$) is 1. Furthermore, the distance between the two nodes without a directed path is $\infty$.

**Definition 10** (Radius of incremental corpus graph). *Define $R$ as a measure of the size of graph.*

All the nodes, whose distance from the initial incremental corpus unit is not greater than a certain value $R$, are added to $V_\iota$, forming the $V_\beta$. When $R = 1$, 3 vertices are included in $V_\beta$ in that example. Obviously,

$$V_\iota \subseteq V_\beta \tag{1}$$

Specifically, $V_\iota = V_\beta$ if $R = 0$.

**Definition 11** (Incremental corpus). *Define $C_\beta$ as a set of incremental corpus units. Actually,*

$$C_\beta = V_\beta \tag{2}$$

Following definitions are relevant to cookbook update.

**Definition 12** (Interval time). *$t_\delta$ defines the update cycle agreed upon by both parties at the algorithm level.*

**Definition 13** (Initial time). *$t_\iota$ defines the moment when communication preparation starts for the first time.*

**Definition 14** (Update start time). *$t_\beta^i$ defines the time when the i-th version of $C_\beta$ starts to build.*

**Definition 15** (Update finish time for sender). *$t_s^i$ defines the moment when the sender completes the codebook update.*

**Definition 16** (Update finish time for receiver). *$t_r^i$ defines the moment when the receiver finishes the codebook update.*

**Definition 17** (Current original corpus). *$C_\alpha^i$ defines the original corpus used between $t_\beta^i$ and $t_\beta^{i+1}$.*

**Definition 18** (Current incremental corpus). *$C_\beta^i$ defines the valid incremental corpus between $t_\beta^i$ and $t_\beta^{i+1}$.*

**Definition 19** (Current synthetic corpus). *$C_\gamma^i$ defines the valid synthetic corpus between $t_\beta^i$ and $t_\beta^{i+1}$.*

**Table 1.** Symbols and Definitions.

| Symbol | Definition |
|---|---|
| $C_\alpha$ | original corpus |
| $C_\gamma$ | synthetic corpus |
| $v_j^0$ | initial incremental corpus unit |
| $v_j^i$ | incremental corpus unit |
| $G_\iota(V_\iota, E_\iota)$ | initial incremental corpus graph |
| $G_\beta\left(V_\beta, E_\beta\right)$ | incremental corpus graph |
| $E_\iota$ | set of initial edges |
| $V_\iota$ | $= \left\{ v_j^0 \middle| j \in \mathbb{N}_+ \right\}$: set of initial vertices |
| $E_\beta$ | set of edges |
| $V_\beta$ | $= \left\{ v_j^i \middle| j \in \mathbb{N}_+, i \in \mathbb{N} \right\}$: set of vertices |
| $d(u, v)$ | distance between node $u$ and $v$ |
| $R$ | $R = \max\limits_{v_j^0 \in V_\iota, v_j^i \in V_\beta} d\left(v_j^0, v_j^i\right)$: radius of incremental corpus graph |
| $C_\beta$ | incremental corpus |
| $D$ | word vector dimension |
| $D'$ | hash vector dimension |
| $X$ | total number of bits of key |
| $N_i$ | part of key |
| $t_\delta$ | interval time |
| $t_\iota$ | initial time |
| $t_\beta^i$ | update start time |
| $t_s^i$ | update finish time for sender |
| $t_r^i$ | update finish time for receiver |
| $C_\alpha^i$ | current original corpus |
| $C_\beta^i$ | current incremental corpus |
| $C_\gamma^i$ | current synthetic corpus |

## 5. Key

The key used in TEDL includes the following components:

$$X = X_1 + X_2 + X_3 + X_4 \tag{3}$$

The meanings of symbols are as follows:

- $X_1$: The $X_1$-bit binary number $N_1$ indicates the initial address. It may be specific to the chapter number or even page number.

- $X_2$: The $X_2$-bit binary number $N_2$ is equal to $R$.

$$R = N_2, 0 \leq N_2 < 2^{X_2} - 1, N_2 \in \mathbb{N} \tag{4}$$

- $X_3$: The $X_3$-bit binary number $N_3$ is used to calculate the dimension $D$ of a word vector. Considering that $D$ is required to be a multiple of 5 in the subsequent process of dealing with them, which will be detailed later, the value range of $D$ is

$$D = 10 + 5N_3, 0 \leq N_3 < 2^{X_3} - 1, N_3 \in \mathbb{N} \tag{5}$$

- $X_4$: The $X_4$-bit binary number $N_4$ is equal to the seed used for initialization of word vectors. For example, the initial vectors for each word $w$ are set with a hash of the concatenation of $w$ and $str(seed)$, where $seed = N_4$.

## 6. Encryption and Decryption

### 6.1. Synthetic Corpus

Both parties build $C_\gamma$ based on the contents of the key ($N_1$ and $N_2$). For different kinds of addresses, the process is similar but slightly different. In the previous example, we have illustrated how ISBN serves as the initial address, which is relatively easy to comprehend. For a better understanding of $C_\gamma$ construction, we take a more complicated example, where arXiv ID is adopted as the address.

Assuming $N_1 = 0001111100000100111001111001012$ and $N_2 = 10_2$, we can find a paper according to arXiv ID *arXiv:1301.03781*, whose content is denoted as $v_1^0$. Besides, it has 32 references denoted as $v_1^1, \cdots, v_{32}^1$. To date, $R = 1$, which does not satisfy $N_2 = R$. Given that each reference cites other. Therefore, we can enlarge the content due to further citations. Each of them is an incremental corpus unit $v_j^i$ and all compose an incremental corpus $C_\beta$. Finally, we add it to the $C_\alpha$ to construct $C_\gamma$, shown in Figure 3.

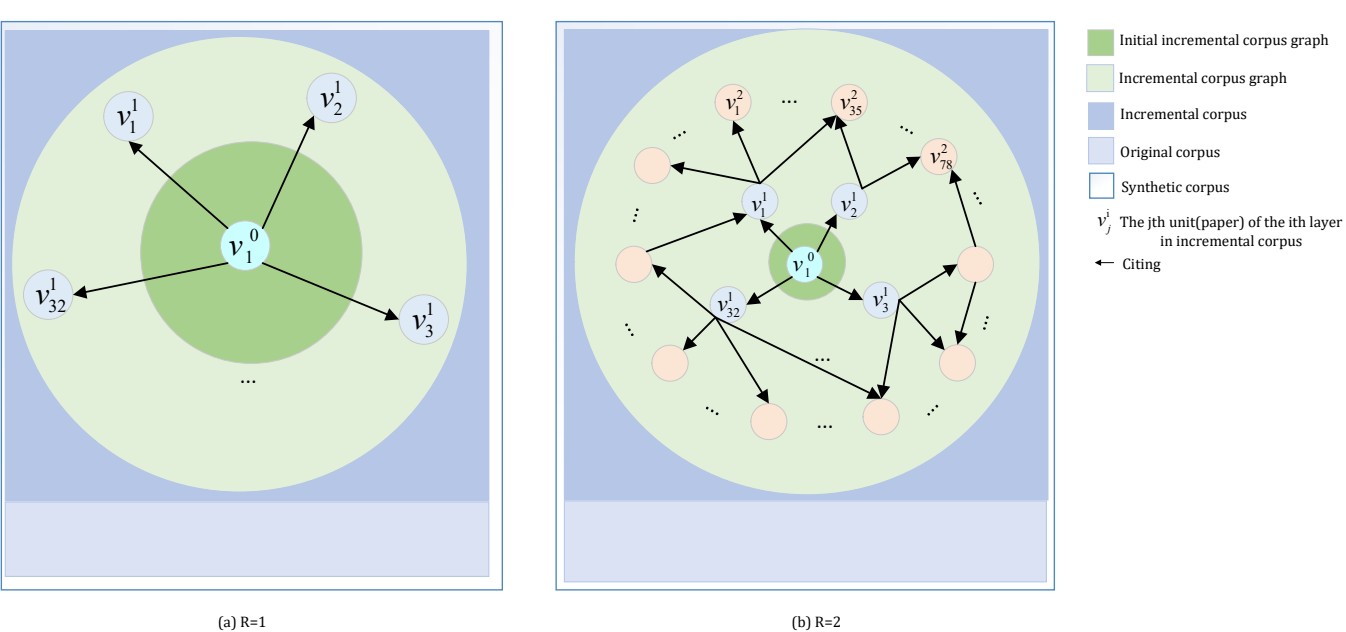

(a) R=1　　　　　　　　　　　　　　　　　　　(b) R=2

**Figure 3.** A more complex example of $C_\gamma$ construction using arXiv ID address.

It is worth mentioning that the language of $C_\beta$ does not require the same as $C_\alpha$, which may work sufficiently well for encryption since we do not need word vectors to have a good performance on the semantic representation.

*6.2. Training*

After obtaining $C_\gamma$, both sides perform the training with deep learning model according to the hyperparameters determined by the $N_3$ and $N_4$.

Firstly, we select a proper model to facilitate the discussion below. It should be qualified for the following Model Requirements:

1. Own at least a public training set.
2. The incremental training set (e.g., $C_\beta$ ) can be addressed with a key and should not be deliberately manufactured but ubiquitous or at least accessible to both parties.
3. The trained parameters should be sufficient and develop some relationship with the data objects.
4. It is more suitable for an unsupervised model or a semi-supervised model. The supervisory part of the latter should be reflected in the public training set. As for a supervised learning model, it is acceptable if it meets Model Requirement 2 after both parties to communications negotiate additional conditions. For example, they agree on a uniform label for the incremental training set.

Obviously, the word embedding model meets those requirements.

Training is the core step in TEDL. In the following, we will discuss what kind of word embedding model is suitable and put forward some precautions in the training process.

6.2.1. Sparse Word Vectors and Dense Word Vectors

Models for word embedding are divided into two categories, namely the sparse word embedding model (e.g.,VSMs [38]) and the dense word embedding model (e.g., Word2vec [15,16]). In the sparse word embedding model, the word-context matrix is constructed, and its initial form is a matrix of frequencies. Each element in a frequency matrix is determined by cooccurrence times of a certain word in a certain context. In practice, the process of matrix construction can be time-consuming when the corpus is large. The entire corpus needs to be scanned, in which each word and its corresponding frequency are recorded, and the results are finally placed in a matrix [39], denoted by **F**. The row vector of the $i$-th row of the word-context frequency matrix corresponds to the word $w_i$, denoted as $\mathbf{f}_{(i:)}$, and the column vector of the $j$-th column corresponds to the context $c_j$, denoted as $\mathbf{f}_{(:j)}$. The value of $f_{ij}$ is expressed as the frequency at which the $i$-th word co-occurs with the $j$-th context. This matrix has $n_r$ rows and $n_c$ columns.

Based on the initial matrix of frequencies, some adjustments are made to weight the elements in the matrix. Ref. [40] has proposed the Pointwise Mutual Information (PMI), which works well for word-context matrics. Furthermore, the variation of PMI, Positive PMI (PPMI) [41], is also a powerful form for distributional representation of words.

When PPMI is applied to **F**, the new matrix, denoted by **X**, has the same size as **F**. The value of an element, denoted by $x_{ij}$, is defined as follows [38]:

$$p_{ij} = \frac{f_{ij}}{\sum_{i=1}^{n^r} \sum_{j=1}^{n^c} f_{ij}} \tag{6}$$

$$pmi_{ij} = \log \left( \frac{p_{ij}}{\left(\sum_{i=1}^{n^r} p_{ij}\right) \cdot \left(\sum_{i=j}^{n^c} p_{ij}\right)} \right) \tag{7}$$

$$x_{ij} = \begin{cases} pmi_{ij}, & \text{if } pmi_{ij} > 0 \\ 0, & \text{otherwise} \end{cases} \tag{8}$$

In this definition, $p_{ij}$ is the probability of co-occurrence of the word $w_i$ and the context $c_i$. Apparently, the matrix **X** is very sparse. Furthermore, when $C_\beta$ is added to $C_\alpha$, the size of both matrix **F** and matrix **X** may change. However, most zeroes remain unchanged, causing the risk of crack increasing, especially when selecting a partial component of the word

vector for encryption. For example, if the original vector of word is $\mathbf{v} = (0\ 0\ 0\ 0.5\ 0.5)$ and the new one is $\mathbf{v}' = (0\ 0\ 0\ 0.25\ 0.75)$, it is extremely dangerous when the first 3 dimensions of the vector are used to replace the word for encryption.

Such a sparse matrix is hence not available for our encryption method, due to the invariance of some elements. So we need to adopt dense word vectors.

### 6.2.2. De-Randomization

For encryption methods, there are many requirements to be met, one of which is that the encryption results should be sufficiently random and unique, that is, the output should be consistent for the same input. Because of the random factors in some models (e.g., the negative sampling strategy [16]), it is possible to generate totally different word vectors under the same hyperparameters. For example, in the negative sampling strategy, only a sample of output vectors, selected by random methods (e.g., the roulette-wheel selection via stochastic acceptance), are updated instead of the whole output vectors, accelerating the training. Therefore, it is suitable for other applications but not for encryption.

Besides, note that for a fully deterministically-reproducible result of running, the model must be limited to a single worker thread, to eliminate ordering jitter from OS thread scheduling. There is no case where the word vectors derived by multi-process accelerated are consistent with ones derived by single-process training.

To sum up, it should be prevented that any random behavior results in different outcomes for the same input. Both sides of the communication cannot encrypt or decrypt when randomness exists. Similarly, when both sides carry out information transmission, the attacker cannot use tricks such as negative sampling and multi-process to speed up a brute-force crack. The reason is that, even if the attacker is currently trying the exact key, the derived codebook mastered by the attacker is inconsistent with the one used for communication. We hence choose the skip-gram hierarchical softmax (SGHS) model as an instance.

### 6.3. Word Vector Table

After training, the word vector table is generated, where the word serves as an index unit whose corresponding value is a $D$-dimensional real vector. The word vector table based on $C_\alpha$ is represented as $T_\alpha$, and one based on $C_\gamma$ is denoted as $T_\gamma$. For a certain word $w$, the corresponding vectors in $T_\alpha$ and $T_\gamma$ are $\mathbf{v}_\alpha|_w$ and $\mathbf{v}_\gamma|_w$, respectively.

Let $\mathbf{v}_\alpha|_w$ and $\mathbf{v}_\gamma|_w$ be row vectors. If $T_\gamma$ is not subsequently processed but directly used for encryption, the similarity between $\mathbf{v}_\alpha|_w$ and $\mathbf{v}_\gamma|_w$ should be low enough, otherwise, it is dangerous. The similarity can be measured by the cosine similarity, which is defined as

$$sim_{xx}(w) = \cos\left(\mathbf{v}_\alpha|_w \cdot \mathbf{v}_\gamma^T\Big|_w\right) \tag{9}$$

The first condition to ensure security is:

$$sim_{xx}(w) < limit_{xx}, \quad 0 < limit_{xx} < 1 \tag{10}$$

where $limit_{xx}$ is a parameter, determined by the security requirements related to the specific application scenario.

In addition, in $T_\alpha$, there exist words with high similarity to $w$. They are usually synonyms of $w$ or words closely related to it. Rank them as $w_1, w_2, w_3, \cdots$ according to the similarity with $w$. The similarity between $w_i$ and $w$ is defined as

$$sim_{xy}(w, w_i) = \cos\left(\mathbf{v}_\alpha|_w \cdot \mathbf{v}_\alpha^T\Big|_{w_i}\right) \tag{11}$$

Obviously, the following relationship is true.

$$sim_{xy}(w, w_{i-1}) > sim_{xy}(w, w_i), \quad i = 2, 3, \cdots \tag{12}$$

Then we give the second condition for ensuring security.

$$sim_{xx}(w) < sim_{xy}(w, w_n), n = limit_{xy} \in \mathbb{N} \tag{13}$$

where $limit_{xy}$ is a parameter. If the $sim_{xx}(w)$ is too large, or even greater than $sim_{xy}(w, w_1)$, the attacker can easily conclude that the plaintext corresponding to the ciphertext $\mathbf{v}_\gamma|_w$ is exactly $w$. Therefore, $limit_{xy}$ should also be set properly according to the security requirements.

In the case where hyperparameters are identical, $sim_{xx}(w)$ is actually determined by the ratio of $C_\beta$ to $C_\alpha$. To meet both requirements for $sim_{xx}(w)$, make sure a proper size of $C_\beta$.

*6.4. Time-Varying Codebook*

Obviously, if the word vector table ($T_\gamma$) is directly used for encryption, the above conditions are not enough for security, calling for more careful and sophisticated design. Nevertheless, it remains difficult to determine whether the design is safe or not. Here we adopt a relatively more trustworthy way: process $T_\gamma$ with SHA-256 function, for its avalanche effect and irreversibility. Although there are some algorithms which are safer and faster than SHA-256, the ciphertext will occupy more space. For example, SHA-512 will occupy space as two times as SHA-256. Therefore, we select SHA-256 on the basis of balancing the security and space performance.

Since Sigmoid function $\sigma(x) = \frac{1}{1+e^{-x}}$ is used in the SGHS model, the derived real vectors are irrational vectors, precisely. In theory, irrational numbers are infinitely long but limited by the computational accuracy of a computer, the results are finite and should be kept as a few effective numbers. To simplify the discussion, double-precision floating-point numbers are specified in the program, which means that the real numbers in $T_\gamma$ are truncated to 16-digit precision or 53-bit precision.

To send the word vector to SHA-256 function, a simple method is to convert the first 16 significant digits of each dimension into a 16-digit integer. For example,

$$0.0006421631111111111_{10} \Rightarrow 6421631111111111_{10}$$

$$6421631111111111112341_{10} \Rightarrow 6421631111111111_{10}$$

If the dimension $D = 200$, all 16-digit integers are spliced to obtain a 3200-digit integer, feeding SHA-256. However, it is extremely time-consuming, degrading the encryption efficiency.

Conversely, feeding a short integer string causes a significant waste of space. For a 32-digit integer, at most $10^{32}$ different results can be generated, far less than $2^{256}$, the space of the message digest generated by SHA-256. Therefore, we consider connecting five 16-digit integers together, the space of which is $10^{80} \approx 864 \times 2^{256}$. Therefore, the transformation for a vector of word $w_i$ is illustrated as Figure 4.

| real | real | ... | real | $D$-dimensional |
|---|---|---|---|---|
| 16-digit | 16-digit | ... | 16-digit | $D$-dimensional |
| 80-digit | 80-digit | ... | 80-digit | $D' = D/5$ $D'$-dimensional |
| 256-bit $h_{i,0}^0$ | 256-bit $h_{i,1}^0$ | ... | 256-bit $h_{i,D'-1}^0$ | $D'$-dimensional |

**Figure 4.** Basic transformation.

As information is transmitted, the primary process of communication is shown in Figure 5, where $D = 200$, $h_{i,j}^0$ denotes the $j$-dimensional in the vector shown as the fourth line in Figure 4. Its corresponding word is $w_i$.

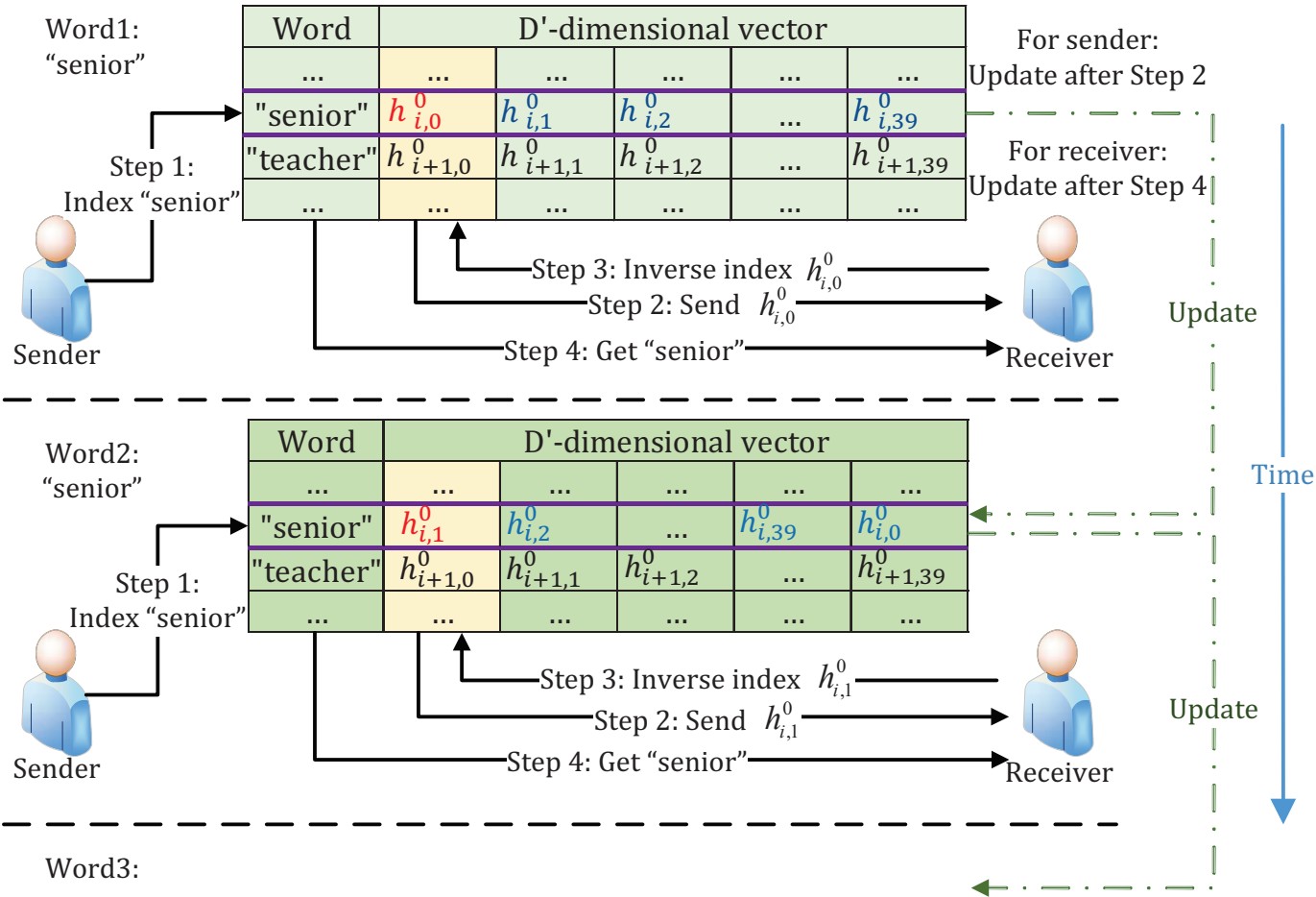

**Figure 5.** Primary processing for word vector table.

In essence, it is similar to the polyalphabetic cipher [42] for resistance to frequency analysis. In other words, in the case where the same word is used multiple times, the corresponding hash, always the first component in a hash vector, is indexed from a different vector table each time.

However, it is not safe enough due to the relatively limited vector tables. Therefore, we divide the $D'$-dimensional vector into two parts, a $(N_3 + 1)$-dimensional vector and a 1-dimensional vector, which are named loop vector and reserved vector, respectively. The more complex and safer processing is shown in Figure 6. $h_{i,j}^k$ denotes a value in loop vectors while $rh_i$ denotes a value in reserved vectors.

$$rh_i = h_{i,D'-1}^0 \tag{14}$$

$$h_{i,j}^k = hash\left(h_{i,j}^{k-1}||rh_i\right), \quad k = 1, 2, \cdots \tag{15}$$

where || stands for concatenating. The concatenation of two 256-bit values results in a 512-bit number. *hash* denotes the SHA-256 function.

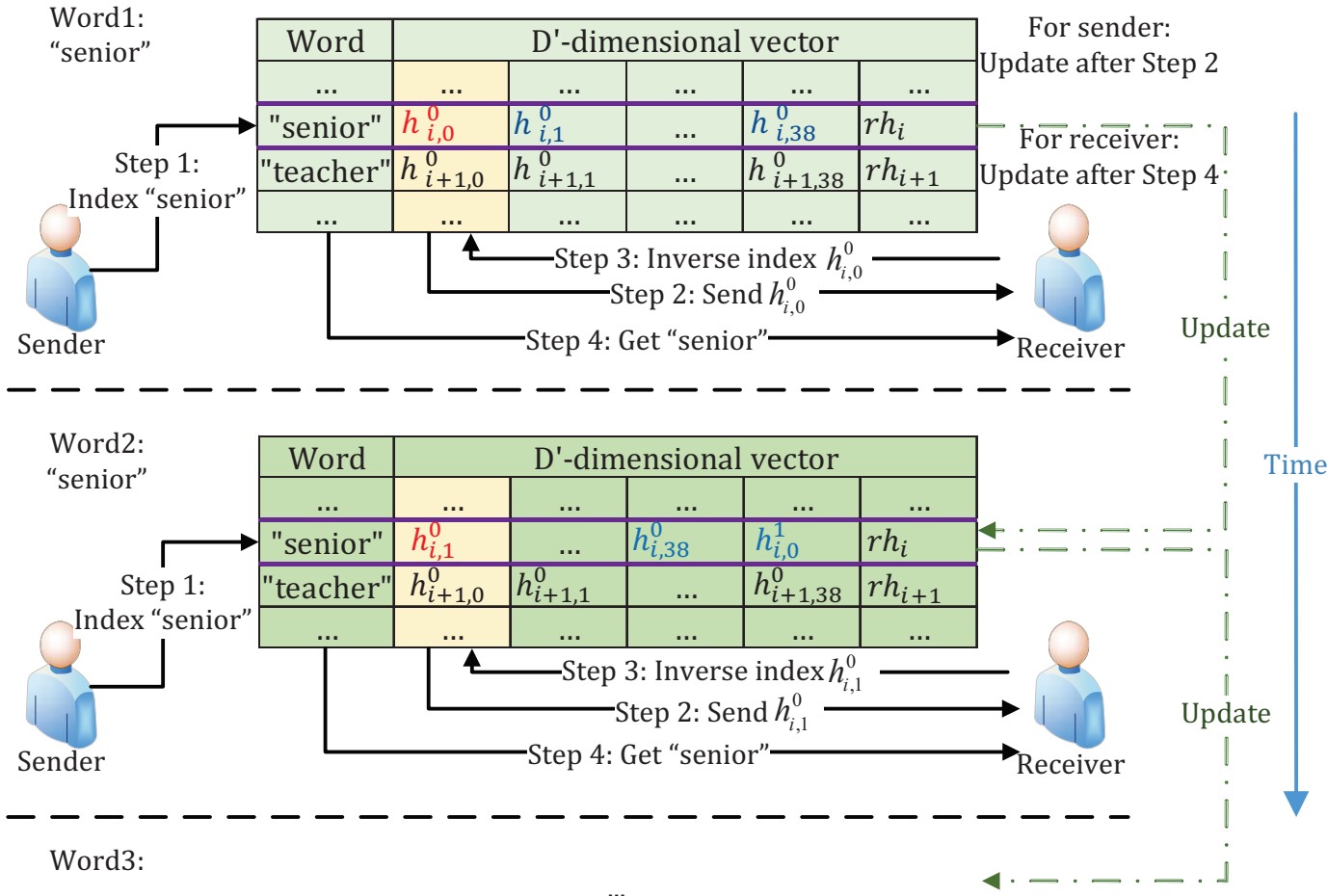

**Figure 6.** Advanced processing for word vector table: codebook.

Each time only the first dimension hash acts as ciphertext. The difference lies in that as the information interacts, the hash vector table keeps changing. Therefore, it is namely a time-varying codebook. Such a design can greatly extend the replacement table. Ideally, since the space of the hash is $2^{256}$, there are $2^{256}$ alternatives to the same word.

In summary, as the two parties communicates, the sender will find the encrypted form of messages in codebook word by word, and replace the encrypted word with the original word to send to the receiver. Furthermore, the receiver will find the decrypted form of messages word by word, decrypting the ciphertext to plaintext. During the communication, the codebook varies in a certain manner, to avoid same plaintexts mapping to same ciphertexts.

## 7. Self-Updating Codebook

The self-updating codebook updates itself periodically without changing the key. To some extent, the time-varying characteristic is a self-renewing mechanism, which is one of the self-update mechanisms of TEDL.

This section explores another self-updating mechanism in TEDL. Considering that the codebook is obtained through a series of steps from $C_\gamma$, the update of the codebook can be achieved by updating $C_\gamma$, or by regulating the hyperparameters (e.g., the seed).

### 7.1. Synthetic Corpus Update

From Definition 12 to 19, $i = 0, 1, \cdots$, and it denotes the $i$-th validity period of the codebook. We make the following reasonable assumptions:

- The time when one gets the key is known to the other.

- From the beginning of constructing $C_\beta$ to the completion of building $C_\gamma$, the content of $C_\beta$ being acquired is static and unchanged.
- Both sender and receiver will not exchange information during the period from the start of the construction of $C_\beta$ to the completion of the codebook update, that is, the communication needs to be aborted from $t_\beta^i$ to $max\left(t_s^i, t_r^i\right)$.

The variables defined above have the following relationship:

$$t_\beta^0 = t_\iota \tag{16}$$

$$t_\beta^i = t_\beta^{i-1} + t_\delta, \qquad\qquad i = 1, 2, \cdots \tag{17}$$

$$C_\alpha^i = C_\gamma^{i-1}, \qquad\qquad i = 1, 2, \cdots \tag{18}$$

$$C_\gamma^i = C_\alpha^i + C_\beta^i, \qquad\qquad i = 0, 1, \cdots \tag{19}$$

Note that, except for $C_\alpha^0$, $C_\alpha^i$ cannot be made public because it is the synthetic corpus $C_\gamma^{i-1}$ in the previous period.

The update process is illustrated as Figure 7. At $t_\beta^0$, both parties begin to construct $C_\beta^0$, adding it to $C_\alpha^0$ to form $C_\gamma^0$. At $t_\beta^1$, $C_\gamma^0$ is renamed to $C_\alpha^1$, which can be further updated to $C_\gamma^1$. Two ways to update are provided here, the choice on which can be negotiated at the algorithm level:

1. Increase $R$ of $G_\beta$, enlarging $C_\beta$.
2. Since there is information transfer between both ends, if the amount of data delivered is sufficiently large, it can act as $C_\beta$.

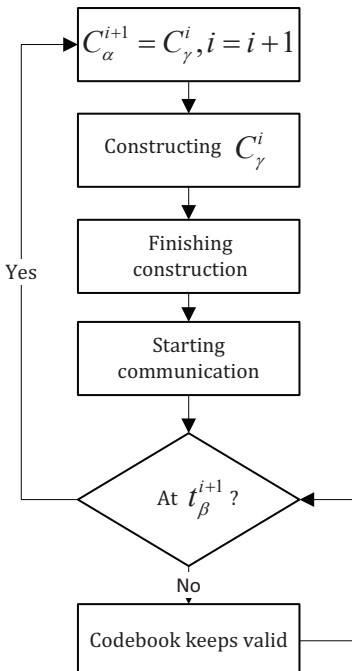

**Figure 7.** Update process.

In theory, infinite rounds of corpus update can be implemented without changing the key. Nevertheless, if the partial deletion is not adopted, the corpus will become larger and larger. Especially, in case of the first way, $C_\beta$ will grow exponentially. If the initial radius $R$ of $G_\beta$ is set to 0, it changes as Figure 8.

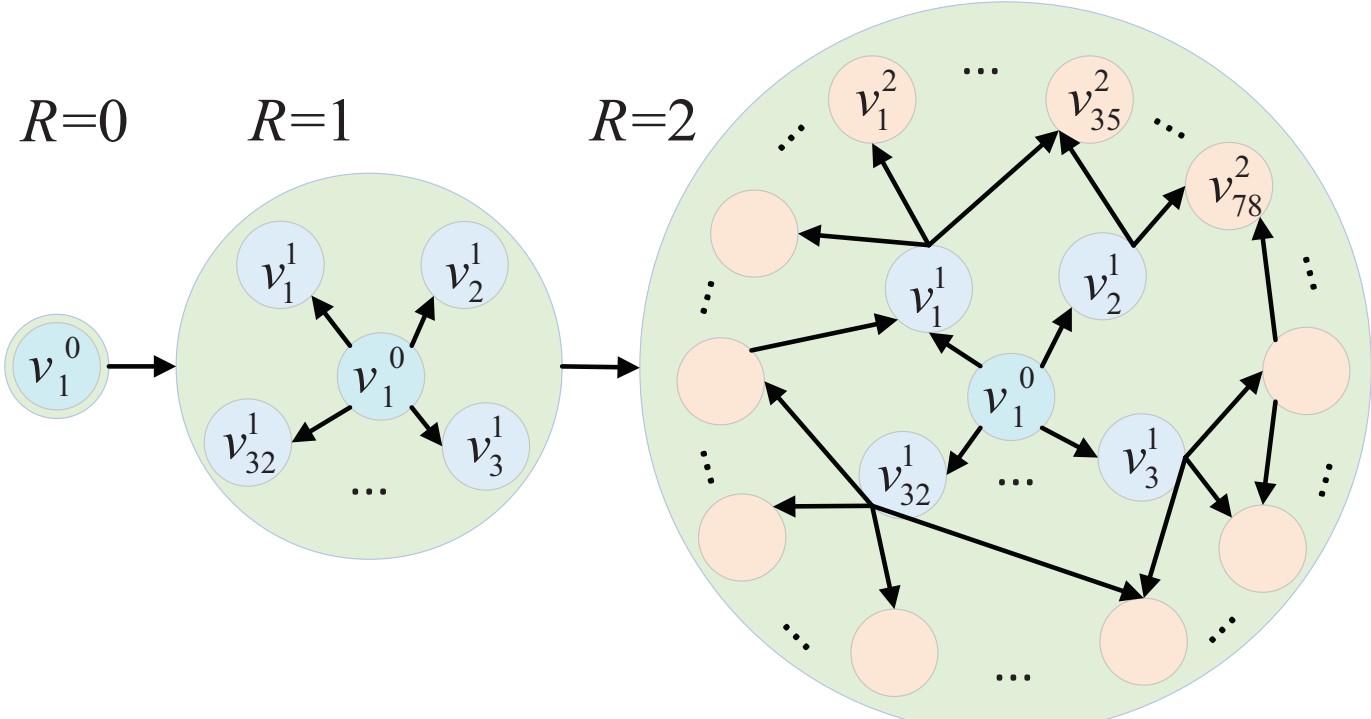

**Figure 8.** The update of $C_\beta$ as $R$ increases.

To meet the conditions suggested in Section 6.3, $C_\beta$ is required correspondingly more. Therefore, it is recommended to agree at the algorithm level that restoration is performed every $x$ times. It means the next version of $C_\alpha^{x-1}$ should be $C_\alpha^0$ instead of $C_\alpha^x$, so Equation (18) is corrected to be

$$C_\alpha^i = \begin{cases} C_\alpha^0, & i = nx \\ C_\gamma^{i-1}, & i \neq nx \end{cases} \quad n, i \in \mathbb{N} \tag{20}$$

In addition to restoration, the split operation is also optional. Assuming that two articles are enough to satisfy the conditions in Section 6.3, we may divide 32 articles into 16 incremental corpora, which are used at $t_\beta^1, t_\beta^2, \cdots, t_\beta^{16}$ in turn. In this way, $C_\gamma$ is controlled to a certain scale by the restore operation and the split operation.

### 7.2. Seed Update

It is also possible to update the codebook by periodically changing the value of the training parameter *seed* and assign it a value from reserved hashes. Two reasons support for choosing a reserved vector:

1.　The value of the reserved vector is constant throughout the encryption and decryption process.
2.　The reserved vector merely serves as partial input of SHA-256 and never exposed. Due to the irreversibility of SHA-256, the attacker cannot derive the value of the reserved vector from the ciphertext.

### 8. Interpretable Word Embedding by Matrix Decomposition

Given that the feasibility of TEDL is based on the fact that the distributed representations of all the words change after adding a small amount of incremental corpus to the original corpus, it is necessary to understand why the training process can achieve the desired effect.

As discribed in Section 6.2, the distributed representations refer to dense word vectors. Two Densification Methods are offered here to generate them instead of sparse vectors.

1. Apply truncated SVD to the sparse matrix derived from a sparse word embedding model.
2. Use a dense word embedding model.

As for Densification Method 1, since the matrix $\mathbf{X}'$ calculated after adding a small amount of corpus can be regarded as the original matrix $\mathbf{X}$ is locally perturbed. If the location of the disturbance is appropriate, the effect is global and each component of word vectors change under finite precision conditions.

As for Densification Method 2, actually, training can be interpreted as matrix decomposition. It has been proved that the embedding process of skip-gram negative sampling (SGNS) and noise-contrastive estimation (NCE) is an implicit matrix decomposition [31], while GloVe [17] is an explicit matrix decomposition [32]. Furthermore, Rong derived and explained the parameter update equations of the Word2vec models [30]. Now, we give a theorem about the essence of the process of skip-gram hierarchical softmax.

**Theorem 1.** *The process of skip-gram hierarchical softmax (SGHS) is an implicit matrix decomposition.*

**Proof.** In a corpus, words $w \in V_w$ and their contexts $c \in V_c$, where $V_w$ and $V_c$ are the word and context vocabularies. The vector representation for word $w_i$ is $\mathbf{v}_{w_i}$, while for $c_i$ is $\mathbf{v}_{c_i}$. For word $w_i$, the contexts are the words surrounding it in an $L$-sized window $w_{i-L}, \cdots, w_{i-1}, w_{i+1}, \cdots, w_{i+L}$, one of which is $c_i$. We denote the collection of observed word-context pairs as $\mathbb{S}$. We use $\#(w_i, c_i)$ to denote the number of times the pair $(w_i, c_i)$ appears in $\mathbb{S}$. Similarly, $\#(w_i)$ is the number of times $w_i$ in $\mathbb{S}$ and $\#(c_i)$ shows the number of times $c_i$ occurred in $\mathbb{S}$. They are defined as:

$$\#(w_i) = \sum_{c_i' \in V_c} \#(w_i, c_i') \tag{21}$$

$$\#(c_i) = \sum_{w_i' \in V_w} \#(w_i', c_i) \tag{22}$$

Consider a word-context pair $(w_i, c_i)$. In the hierarchical softmax model, no output vector representation exists for context words. In other words, the vector $\mathbf{v}_{c_i}$ is untrained. Instead, there is an an output vector $\mathbf{v}'_{n(c_i,j)}$, which is trained during the training process, for each of the $|V_c| - 1$ inner units. Furthermore, the probability of a word being context $c_i$, the output word, is defined as

$$P(w_{out} = c_i \mid w_i) = \prod_{j=1}^{L(c_i)-1} \sigma\left( [\![c_i, j]\!] \cdot \mathbf{v}'^{T}_{n(c_i,j)} \mathbf{v}_{w_i} \right) \tag{23}$$

$$\sigma(x) = \frac{1}{1 + e^{-x}} \tag{24}$$

$$[\![c_i, j]\!] = [\![n(c_i, j+1) = ch(n(c_i, j))]\!] \tag{25}$$

where $L(c_i)$ denotes the length of path, $n(c_i, j)$ means the $j$-th unit on the path from root $w_i$ to the word $c_i$, $ch(n)$ is the left child of unit $n$, $\mathbf{v}'_{n(c_i,j)}$ is the output vector of $n(c_i, j)$, $\mathbf{v}_{w_i}$ is the distribution representation of $w_i$, as well as the output value of the hidden layer, $[\![x]\!]$ is a specially defined function expressed as

$$[\![x]\!] = \begin{cases} 1, & \text{if x is true} \\ -1, & \text{otherwise} \end{cases} \tag{26}$$

Obviously, the following equation is true.

$$\sum_{i=1}^{|V_c|} P(w_{out} = c_i \mid w_i) = 1 \tag{27}$$

The probability of going left at an inner unit (including the root unit) $n$ is defined as

$$P(n, left) = \sigma\left(\mathbf{v}'_n{}^T \cdot \mathbf{v}_{w_i}\right) \tag{28}$$

which is determined by both the output vector of the inner unit and the hidden layer. Similarly, the probability of going from unit $n$ to right is

$$P(n, right) = 1 - \sigma\left(\mathbf{v}'_n{}^T \cdot \mathbf{v}_{w_i}\right) = \sigma\left(-\mathbf{v}'_n{}^T \cdot \mathbf{v}_{w_i}\right) \tag{29}$$

The parameter update process is derived in the following part. For simplicity, we consider the situation of one-word context models, which are easily extended to skip-gram models. We simplify some notations without introducing ambiguity first:

$$[\![\cdot]\!] = [\![n(c_i, j+1) = ch(n(c_i, j))]\!] \tag{30}$$

$$\mathbf{v}'_j = \mathbf{v}'_{n(c_i, j)} \tag{31}$$

The global objective is trained using stochastic gradient updates over the observed pairs in $\mathbb{S}$, defined as

$$E = \sum_{w \in V_w} \sum_{n \in V_c} \#(w, c) \log P(w_{out} = c \mid w) \tag{32}$$

For a training instance, the local objective is defined as

$$E(w_i, c_i) = \log P(w_{out} = c \mid w) \tag{33}$$

$$= \sum_{j=1}^{L(c_i)-1} \log \sigma\left([\![\cdot]\!] \mathbf{v}'_j{}^T \mathbf{v}_{w_i}\right) \tag{34}$$

In SGHS, it is the vectors of the inner units and a hidden layer that are trained. On the path from $w_i$ to $c_i$, for each inner unit, $[\![\cdot]\!]$ is either 1 or $-1$. Furthermore, for each $w_i$, there exist $\#(w_i)$ paths (including cases where the same path is repeatedly counted). Assume that a total of $k$ paths pass through $n$, there must be $k_l$ paths through the left child nodes of $n$, while $k_r$ paths walk via its right child, obtaining:

$$k = k_l + k_r \leq \#(w_i) \tag{35}$$

The global objective hence is rewritten as

$$E = \sum_{w_i \in V_w} \sum_{c_i \in V_c} \#(w_i, c_i) \sum_{j=1}^{L(c_i)-1} \log \sigma\left([\![\cdot]\!] \mathbf{v}'_j{}^T \mathbf{v}_{w_i}\right) \tag{36}$$

$$= \sum_{w_i \in V_w} \sum_{n \in V_n} \left[ k_r + (k_l - k_r) \log \sigma\left(\mathbf{v}'_n{}^T \mathbf{v}_{w_i}\right) \right] \tag{37}$$

where $V_n$ denotes the collection of inner units $n$. We take the derivative of $E$ with regard to $\mathbf{v}'_n{}^T \mathbf{v}_{w_i}$, obtaining

$$\frac{\partial E}{\partial \mathbf{v}'_n{}^T \mathbf{v}_{w_i}} = \sum_{w_i \in V_w} \sum_{n \in V_n} \left[ k_l - (k_l + k_r) \sigma \left( \mathbf{v}'_n{}^T \mathbf{v}_{w_i} \right) \right] \tag{38}$$

We compare the derivative to zero, arriving at

$$\sigma \left( \mathbf{v}'_n{}^T \mathbf{v}_{w_i} \right) = \frac{k_l}{k_l + k_r} \tag{39}$$

Hence,

$$\mathbf{v}'_n{}^T \mathbf{v}_{w_i} = \ln k_l - \ln k_r \tag{40}$$

Finally, we can describe the matrix $\mathbf{M}$ of $|V_w|$ rows and $|V_n|$ columns that SGHS is factorizing:

$$\mathbf{M}_{in}^{SGHS} = \mathbf{v}'_n{}^T \mathbf{v}_{w_i} = \ln k_l - \ln k_r \tag{41}$$

Therefore, the embedding process of SGHS also performs an implicit matrix decomposition. □

Subtle modification to the original corpus is equivalent to perturbation on the implicit matrix, which can eventually lead to radical change in the training results.

However, just matrix decomposition, whether explicit or implicit, cannot ensure that every dimension of each word vector changes. Very few individual components of the word vector may remain unchanged, which is a fatal weakness for encryption. From this perspective, Densification Method 1 is not secure enough. Instead, Densification Method 2 can solve this problem by increasing the number of iteration epoch. In particular, as long as it is greater than one round, the effect of local disturbances will be comprehensive, resulting in changes in all word vectors, which is confirmed by related experiments.

## 9. Security Analysis

According to the Kerckhoff guidelines, a good encryption method should have a large enough key space. The key space of TEDL is $2^X$. Considering the example with arXiv address, set $X_1 = 30$, $X_2 = 2$, $X_3 = 8$, $X_4 = 256$. In theory, $X_4$ can be infinite, but given that the space of hash is $2^{256}$, we assign that $X_4 = 256$. Otherwise, referring to the *drawer principle*, there must be two different hashes colliding. See Table 2 for a comparison of the key space.

For the same key space, the time required to complete encryption for each key differs. There may be doubts here: for the attacker, the longer time for trying each key, the more time will be spent on the crack, but in turn, does the time required for the communication parties to normally transmit information increase dramatically? It is not the case for TEDL, owing to the two-stage structure. The brute-force crack is mainly performed at the first stage while the communication between the two parties is mainly carried out at the second stage. Therefore, it improves safety while ensuring efficiency.

In addition, TEDL does not directly use the key in the encryption. Instead, the key is the instructor during the encryption and decryption process. Therefore, techniques that involve ciphertexts analysis, such as Differential Cryptanalysis [43], Linear Cryptanalysis [44], Truncated Differentials [45], Boomerang Attacks [46], Impossible Differentials [47] and others [48,49], are not effective since these ciphertexts involve limited knowledge about keys, making it infeasible for attackers to predict keys.

Besides, TEDL bases the security on the difficulty in parameter interpretation in deep learning, which is another hard problem. Not only the parameters themselves are uninterpretable, but the trend of their variation is also unexplained, which is core challenge, as described in [50].

**Table 2.** Key space.

| Method | Key Bits | Key Space |
| --- | --- | --- |
| TEDL | 296 | $2^{296}$ |
| AES | 256 | $2^{256}$ |
| Salsa20 | 256 | $2^{256}$ |
| 3DES | 168 | $2^{168}$ |
| DES | 56 | $2^{56}$ |

Furthermore, the application of SHA-256 makes TEDL more secure. In cryptography, the avalanche effect refers to an ideal property: when the input makes the slightest change (for example, inverting a binary bit), an indistinguishable change in the output occurs (there is a 50% probability that each binary bit in the output is inverted). The ideal state of nonlinear diffusivity is the avalanche effect. Ref. [51] shows that SHA-256 has excellent nonlinear diffusivity. Therefore, even if word vectors are similar, the outputs are quite different. Moreover, because of the irreversibility of the secure hash function, the relation between ciphertext and plaintext is extremely weak and intractable. It is impossible for an attacker to decrypt. Finally, other related security analyses will be illustrated with experimental results.

## 10. Experiments and Performance Analysis

This section presents the experiments and corresponding analysis of TEDL, showing that it achieves a balance between security and efficiency, which make it suitable for transmission of a large amount of data. All the experiments are performed on an identical platform with system configuration of i7 processor @ 2.50 GHz and 8 GB Ram, and evaluated on two datasets in different languages, one is Chinese Wikipedia corpus (about 1.3 GB) (https://dumps.wikimedia.org/zhwiki/) and the other is a subset of the English Wikipedia corpus (about 600 MB) (https://dumps.wikimedia.org/enwiki/). Relative codes are available on GitHub (https://github.com/AmbitionXiang/TEDL).

Our experiments focus on following issues:

- *Recovery*. It verifies that the ciphertext is decrypted successfully, even if the ciphertext is tampered with in an insecure channel.
- *Consumed time for brute force*. It measures the ability of encryption methods to resist brute force attacks.
- *Frequency analysis*. It is about the frequency distribution of cipher symbols, characterizes the confusion.
- *Correlation*. It refers to the correlation analysis between encrypted data and original data.
- *Sensitivity analysis*. It also measures the strength of encryption methods against cracking and hacking threats. For plaintext sensitivity or key sensitivity, it is high when changing a small number of bits in plaintext or key results in a large variance in ciphertext. As for ciphertext sensitivity, it is embodied when a natural error or intentional tampering in the ciphertext is remarkable [52].
- *Efficiency analysis*. It measures encryption speed.
- *Generality analysis*. It studies whether a method is suitable for multiple models.

### 10.1. Recovery

As described in Section 6.4, both parties only send and receive the first-dimension hash in the time-varying codebook. The hash that can be restored to a word is called a valid hash. Obviously, the number of valid hashes is equal to the total number of word keys in the codebook, far less than $2^{256}$. When the ciphertext is partially tampered with, the valid hash that has the most overlap with it is selected from all the hashes of the first dimension,

the plaintext hence can be restored with a high probability. We define the recovery accuracy rate (RACR) to measure the anti-interference and recoverability of TEDL.

$$RACR = \frac{n_c}{n_t} \tag{42}$$

where $n_t$ denotes the total number of tampered hashes, $n_c$ stands for the count of tampered hashes which are successfully restored to correct words.

　　We randomly select 1000 words as samples, whose corresponding hashes are tampered with. To test the recovery rate under different conditions, we set the total number of tampered bits from 0 to 256 bits, and randomly choose the tampering location. Experiments repeat to calculate RACR and we plot it versus the number of tampered bits as Figure 9.

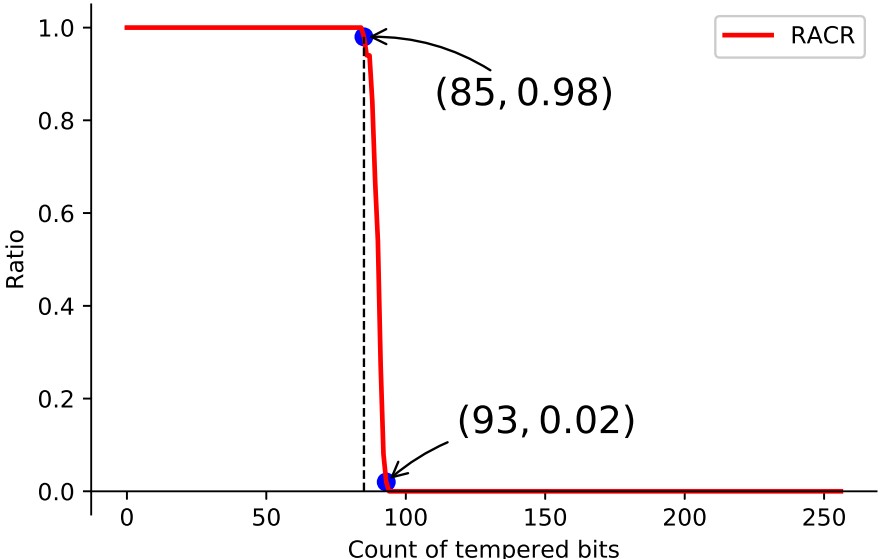

**Figure 9.** Recoverability of tampered ciphertexts.

　　It shows that an invalid hash is recoverable if the count of tampered bits is less than 85, otherwise, the original word is not able to be restored.

### 10.2. Consumed Time for Brute Force

　　Now we explore the relationship between the time spent in stage one and some of the training parameters (the number of iterations, the dimension $D$) for an identical synthetic corpus (about 1 GB). The time of stage one consists of two parts, time on model training and processing the original word vector table. The experimental results are drawn as Figure 10. Obviously, time cost of brute force is increasing linearly with the number of dimensions and iterations.

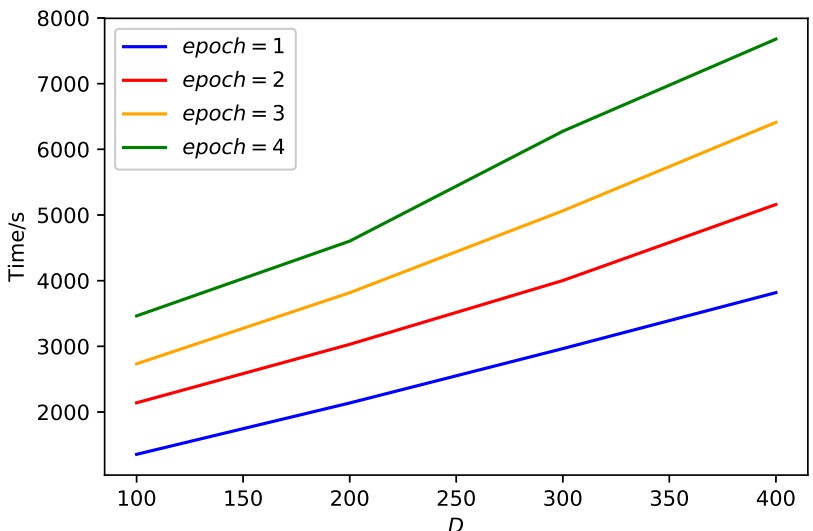

**Figure 10.** Time cost on stage one.

### 10.3. Frequency Analysis

For English text, we explore texts of 2, 20, and 200 MB respectively, which are extracted from corpus (https://dumps.wikimedia.org/enwiki/), and plot frequency distribution histograms with respect to both plaintext and ciphertext, which are shown in Figure 11a.

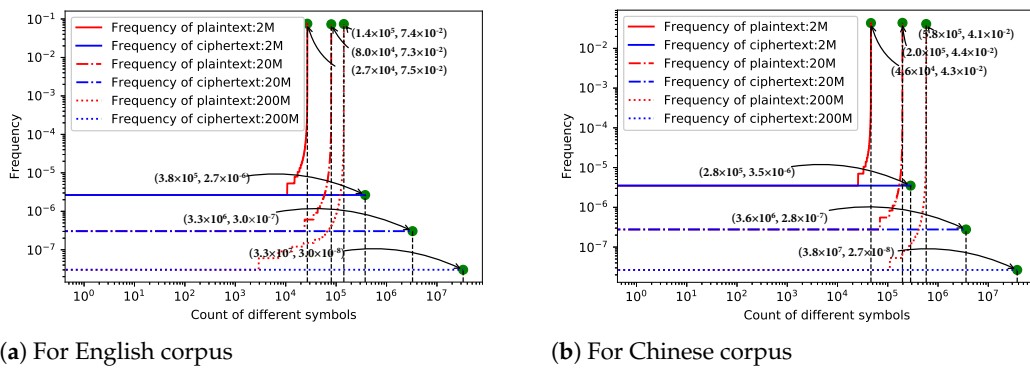

(**a**) For English corpus         (**b**) For Chinese corpus

**Figure 11.** Frequency distribution of plaintext and ciphertext.

For text in Chinese, 2, 20 and 200MB texts are encrypted accordingly, and the frequency histogram is shown in Figure 11b. Obviously, TEDL completely dissipated the original distribution instead by a fairly uniform distribution. The distribution more uniform, the more security it guarantees. As it is uniform, one cannot infer the plaintext from the frequency distribution. It has a remarkable ability to resist against the statistical attack, especially frequency analysis, and works in any language.

### 10.4. Correlation

The original data is instinctively considered words, while the encrypted data is either word vectors or 256-bit hashes. It is hard to calculate the correlation. Therefore, we need to redefine original data and encrypted data.

1. In the case of directly using the original word vector table, original data is denoted by $\mathbf{v}_\alpha|_w$, which is generated after embedding training on the public corpus under certain parameters, while encrypted data is $\mathbf{v}_\gamma|_w$.

2.   In the case of using a time-varying codebook, $h_\alpha$ represents original data and $h_\gamma$ denotes encrypted data. Both is obtained by further processing described as Section 6.4, and they are corresponding to $\mathbf{v}_\alpha|_w$ and $\mathbf{v}_\gamma|_{w'}$, respectively.

We carry out experiments in both cases mentioned above.

### 10.4.1. Directly Using Word Vector Table

In case 1, the correlation between the encrypted data and the original data is measured by the cosine similarity defined by Equation (9), rewritten as follows:

$$sim_{xx}(w) = \cos\left( \mathbf{v}_\alpha|_w \cdot \mathbf{v}_\gamma^T\big|_w \right) \tag{43}$$

We explore the effects of different training conditions, including the number of iterations (*epoch*), the vector dimension (*D*), the ratio of $C_\beta$ to $C_\alpha$ (*C ratio*) and the size of window (*window*), which indicates the maximum distance between the current and predicted word within a sentence in the word embedding model.

### Changing **epoch** and **D**

For other parameters, we set *C ratio* = 1:10,000, *seed* = 1, *window* = 5. We sort the *sim* of each word from small to large and draw as Figure 12. Obviously, as *D* and *epoch* increase, the correlation between the original data and the encrypted data decreases. Note that in the case where *epoch* = 1, regardless of the word vector dimension, there is a phenomenon that the correlation is 100%. On the other hand, as long as *epoch* is greater than 1, this phenomenon no longer exists. In addition, considering that as *epoch* increases, the training process converges, which means word vectors update slightly. Therefore, the setting of epoch is not included in the key but should be agreed at the algorithm level.

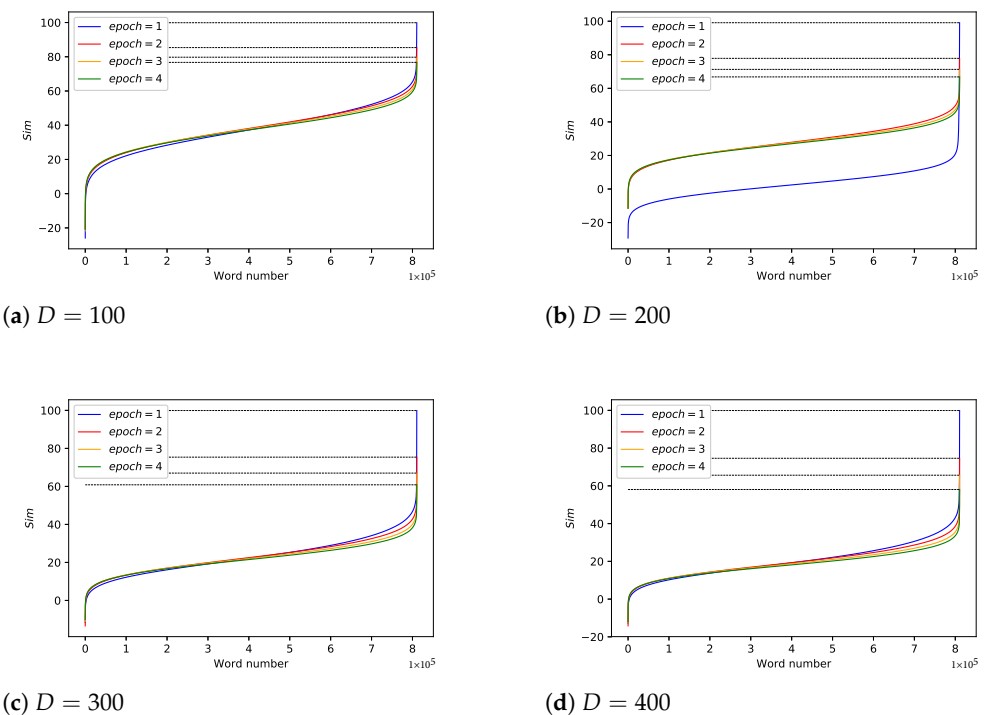

**Figure 12.** Correlation (measured by *sim*) as *D* and *epoch* change.

### Changing **C ratio**

For other parameters, *epoch* = 2, *D* = 200, *seed* = 1, *window* = 5 and we test the correlation at a ratio of 1:10,000, 1:1000, 1:100. The results are shown in Figure 13a. As *C ratio* increases, the *sim* decreases, which is in line with expectations. However, the result

relies on language consistency in $C_\beta$ and $C_\alpha$. If not, will the encryption effect drop? We answer this question in the next experiment. If not specified, subsequent experiments also proceed under the conditions mentioned above.

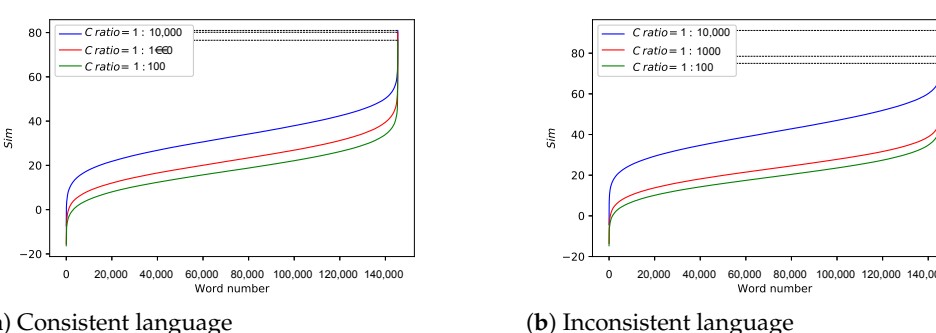

(**a**) Consistent language         (**b**) Inconsistent language

**Figure 13.** Correlation (measured by *sim*) as *C ratio* changes.

Language Inconsistency Exists

Unlike previous experiments, pure Chinese corpus serves as $C_\beta$. Furthermore, we change the *C ratio* to observe trends in correlation, pictured as Figure 13b. Obviously, the encryption effect is not much different from the last result, which indicates the impact of language inconsistency is negligible.

Changing **Window**

Apart from the default conditions, we set *C ratio* = 1:10,000 and depict the result as Figure 14a.

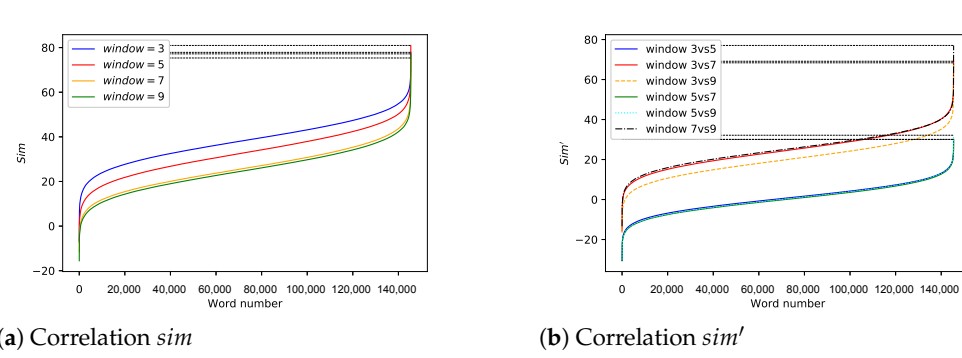

(**a**) Correlation *sim*         (**b**) Correlation *sim'*

**Figure 14.** Correlation trend as *window* changes.

We can conclude that *window* is also related to correlation. However, it does not mean that the variation of *window* directly changes the representation of the same word significantly, while it is evident when changing $D$. Therefore, it is necessary to verify this, under the condition that no $C_\beta$ is added to $C_\alpha$, as well as the default conditions. The Equation (43) is revised as

$$sim'_{xx}(w) = \cos\left(\mathbf{v}_\alpha|_w \cdot \mathbf{v}'_\alpha{}^T\Big|_w\right) \tag{44}$$

where $\mathbf{v}_\alpha$ and $\mathbf{v}'_\alpha$ is trained on the same $C_\alpha$ but with a different parameter *window*. Figure 14b shows the result. We can see the size of *window* directly influences the representations of words. It is feasible to add it to the key content.

### 10.4.2. Using Time-Varying Codebook

In this case, each word is encrypted to a 256-bit hash $h_\gamma$. We conduct experiments on both English and Chinese corpus. As described in [53], the correlation is measured by

$$r_{xy} = \frac{count(h_\alpha \oplus h_\gamma)}{length(h_\alpha)} \tag{45}$$

where $\oplus$ denotes XORing, $h_\alpha \oplus h_\gamma$ generates a binary string, $count(string)$ is the count of '0' in the $string$, $length(h_\alpha)$ represents the length of string $h_\alpha$, which is considered the original data.

As can be seen from the experimental results in Section 10.4.1, as long as $epoch > 1$, each word vector definitely changes. Without losing the generality of test, we confine $epoch = 2$ and test $r_{xy}$ by changing $D$, depicting the frequency distribution histogram (hist) and frequency distribution function (pdf) of $r_{xy}$ as Figure 15.

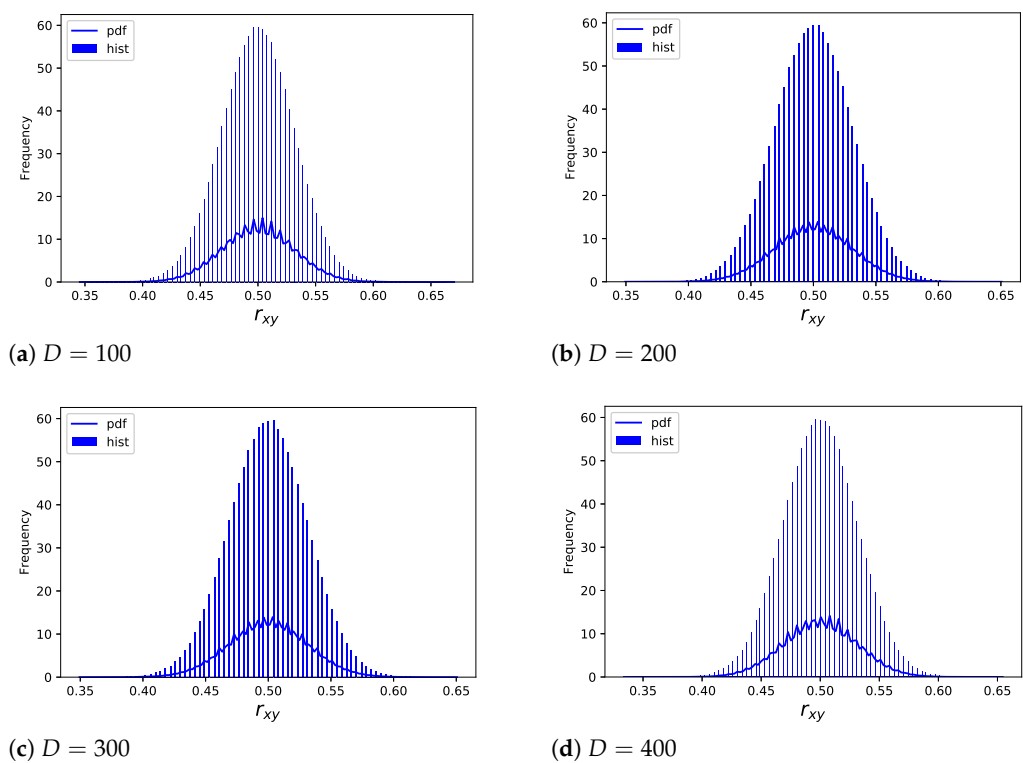

(**a**) $D = 100$

(**b**) $D = 200$

(**c**) $D = 300$

(**d**) $D = 400$

**Figure 15.** Correlation (measured by $r_{xy}$) as $D$ changes.

The horizontal coordinate represents $r_{xy}$, and the vertical coordinate denotes the corresponding frequency. Obviously, $r_{xy}$ is concentrated around 0.5, which means that half of the bits of the encrypted data are changed compared with the original data. It is a good encryption property. Since it is safer to use time-varying codebook, we only consider this case in all subsequent experiments.

### 10.5. Sensitivity Analysis

To measure the sensitivity of plaintext and key, the operation is to make a slight change to either and calculate the change rate of ciphertext (CRC), defined as

$$CRC = \frac{Dif\left(h_\gamma, h'_\gamma\right)}{length(h_\gamma)} \tag{46}$$

where $h_\gamma$ is the original ciphertext, $h'_\gamma$ is the ciphertext as minor modifications occur to plaintext or key, $Dif\left(h_\gamma, h'_\gamma\right)$ is the count of distinct symbols in $h_\gamma$ and $h'_\gamma$.

For ciphertext sensitivity, we disturb several bits of ciphertext to observe whether it is still in the valid hash collection $\mathbb{H}_v$, which contains all valid hashes at the moment. The ciphertext change sensitivity (CCS) is defined as

$$CCS = 1 - \frac{n_{co}}{n_t} \tag{47}$$

where $n_t$ denotes the total number of invalid hashes, $n_{co}$ stands for the number of tampered hashes still in valid hash collection.

### 10.5.1. Key Sensitivity

For a given key, we choose a key that differs by only one bit, which can be located at any component of the key, and juxtapose ciphertexts for the same word. Given that components $N_2$ and $N_3$ are related to the $C$ *ratio* and $D$, respectively, which have already been considered, only $N_1$ and $N_4$ remain to be altered for further experiments.

**Modifying $N_1$**

We transform $N_1 = 00\ 0111\ 1100\ 0001\ 0011\ 1001\ 1110\ 01\mathbf{01}_2$ to $N'_1 = 00\ 0111\ 1100\ 0001\ 0011\ 1001\ 1110\ 01\mathbf{11}_2$ in the example using arXiv ID, which means the address is converted to *arXiv:1301.03783*. Therefore, another paper serves as the whole $C_\beta$ if $N_2 = 0$. Obviously, as $N_2$ grows, disturbing $N_1$ causes a greater impact. For all the words in $C_\alpha$, the changes in their representations are presented in Figure 16a.

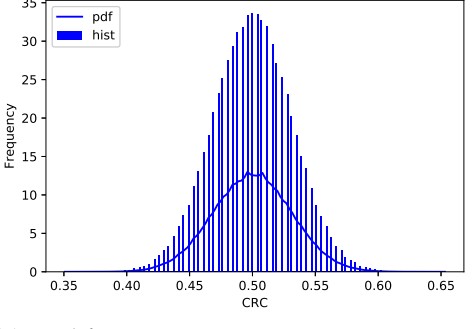
(**a**) Modifying $N_1$

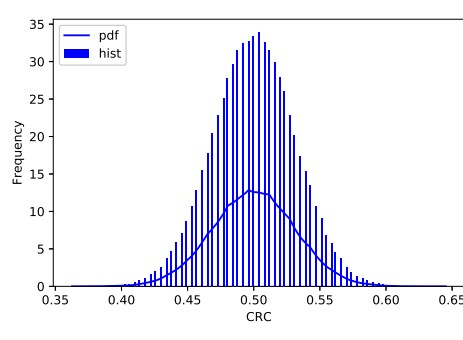
(**b**) Modifying $N_4$

**Figure 16.** Key sensitivity distribution.

**Modifying $N_4$**

Once $N_4$, related to the seed, is altered, the whole initial word vectors will experience the earthquake, resulting in entirely distinct representations. When the seed is changed from 1 to 2, the result is shown in Figure 16b.

From the above two subsections, we know a bit of interference in the key can cause a huge difference in representations. Almost 50% bits in ciphertext reverse, close to the avalanche effect.

### 10.5.2. Plaintext sensitivity

For the word embedding model, words here are atomic. A slight change in plaintext can be interpreted as replacing a primitive word with a synonym or a morphologically similar word. Collins dictionary (https://www.collinsdictionary.com/) is used for searching synonyms.

We select six groups of more common words to experiment, namely "people", "male", "female", "beautiful", "good", "look" and their synonyms, partially shown in Table 3.

**Table 3.** Words and their synonyms.

| Words | Corresponding Synonyms |
|---|---|
| people | persons/humans/individuals/folk/human beings/ humanity/mankind/mortals/the human race |
| female | woman/girl/lady/lass/shelia/charlie/chook/wahine |
| male | masculine/manly/macho/virile/manlike/manful |
| . . . | . . . |

For each word in the first column, we compare the representations of themselves and ones of their synonyms and calculate the CRC. The frequency distribution of which is shown in Figure 17.

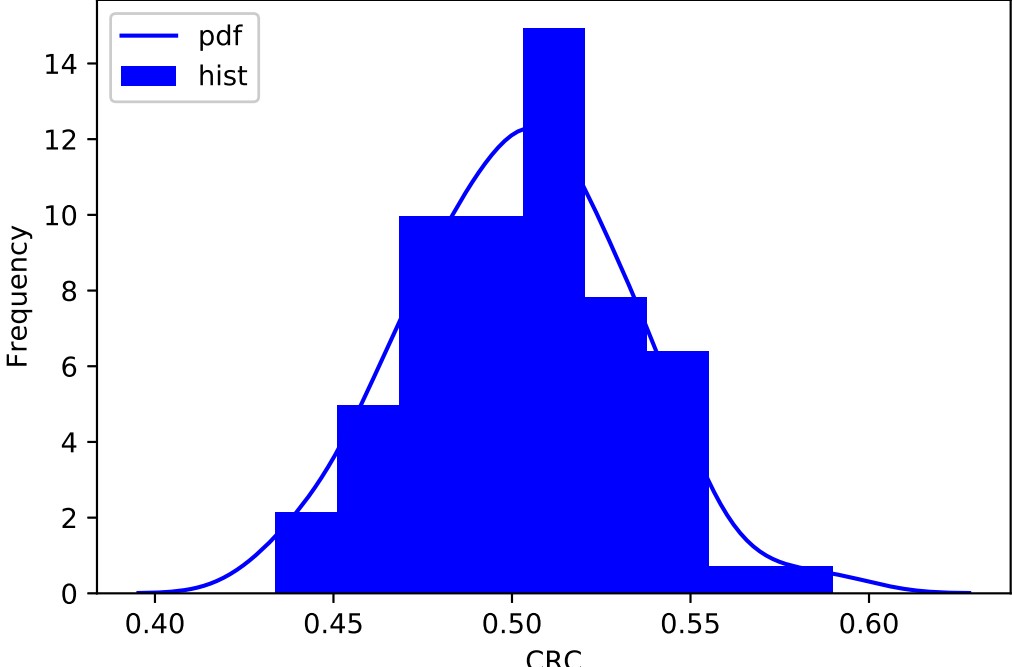

**Figure 17.** Plaintext sensitivity distribution.

Obviously, the distribution of CRC is still concentrated around 50%, that is, it characterizes good sensitivity.

### 10.5.3. Ciphertext Sensitivity

The experimental setup is similar to Section 10.1. We randomly select 1000 words as samples for the experiment, whose corresponding hashes are tampered with. To test CCS under different conditions, we set the total number of tampered bits from 0 to 256 bits, randomly choose the tampering location, and check whether it is still in the valid hash collection. Our experimental result is $CCS \approx 1$ no matter how many bits are tampered with.

As described in Section 6.4, although each word may have $2^{256}$ representations, at a certain moment during the communication, the size of the valid hash collection is equal to the number of different words in $C_\gamma$. The probability is expressed by

$$P(tamper(h_\gamma) \in \mathbb{H}_v) = \frac{|V_w|}{2^{256}} \tag{48}$$

Assuming $|V_w| = 10^8$, $P(tamper(h_\gamma) \in \mathbb{H}_v)$ is approximately equal to 1, which is in accordance with the experimental result.

### 10.6. Efficiency Analysis

The comparison of efficiency among TEDL and some popular cryptosystems is depicted as Figure 18.

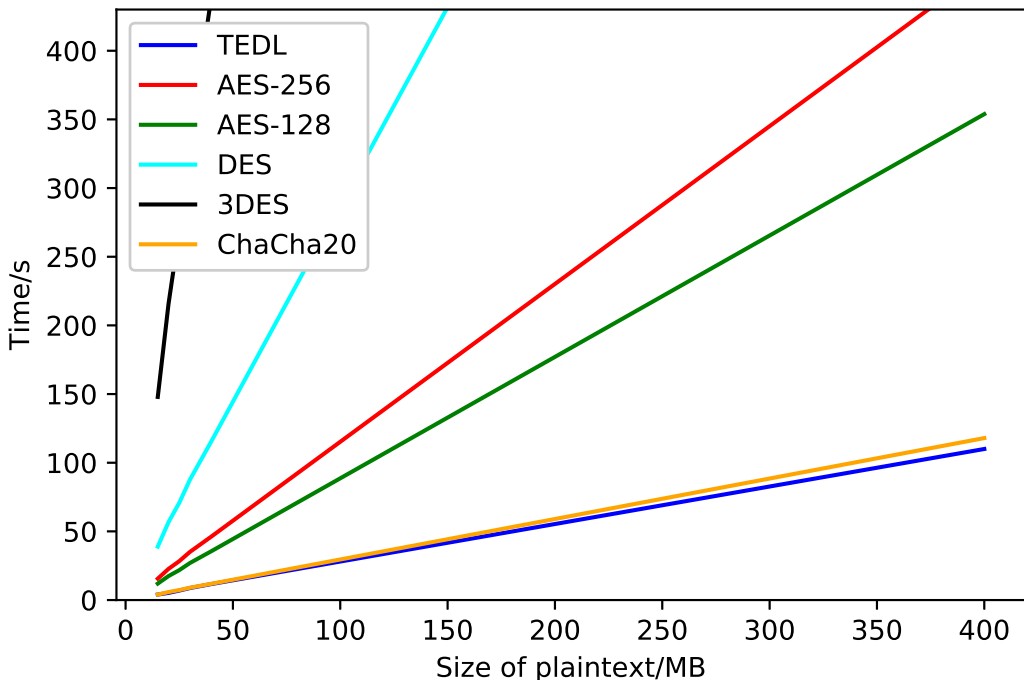

**Figure 18.** Efficiency comparison among encryption methods.

It can be seen that for the same amount of plaintext, TEDL needs least time, which means that TEDL is fast, in other words, it shows that our method is efficient.

### 10.7. Generality Analysis

Multiple word embedding models can be applied in TEDL. In addition to the *Word2vec* used in previous sections, *NNLM* [54], *fastText* [18], and *GloVe* [17] are applied in this section.

Different models have different parameter settings. For example, as for *fastText*, most parameters function the same with those in *Word2vec*. These parameters are set as the default conditions mentioned above, that is $epoch = 2$, $D = 200$, $seed = 1$, $window = 5$ and $C\ ratio$ = 1:10,000. Besides, there exist some unique parameters in *fastText* due to using enriches word vectors with subword(n-grams) information (e.g., the max length of char ngrams as well as the minimum. Here we set them to 5 and 3, respectively). The parameters in other models are set according to the characteristics of the model. However, make sure to de-randomize the training process.

The correlation defined as Equation (45) serves as a representative indicator to measure the effect of encryption, which is shown in Figure 19. It shows that multiple models can be used to encryption, and they behave similarly.

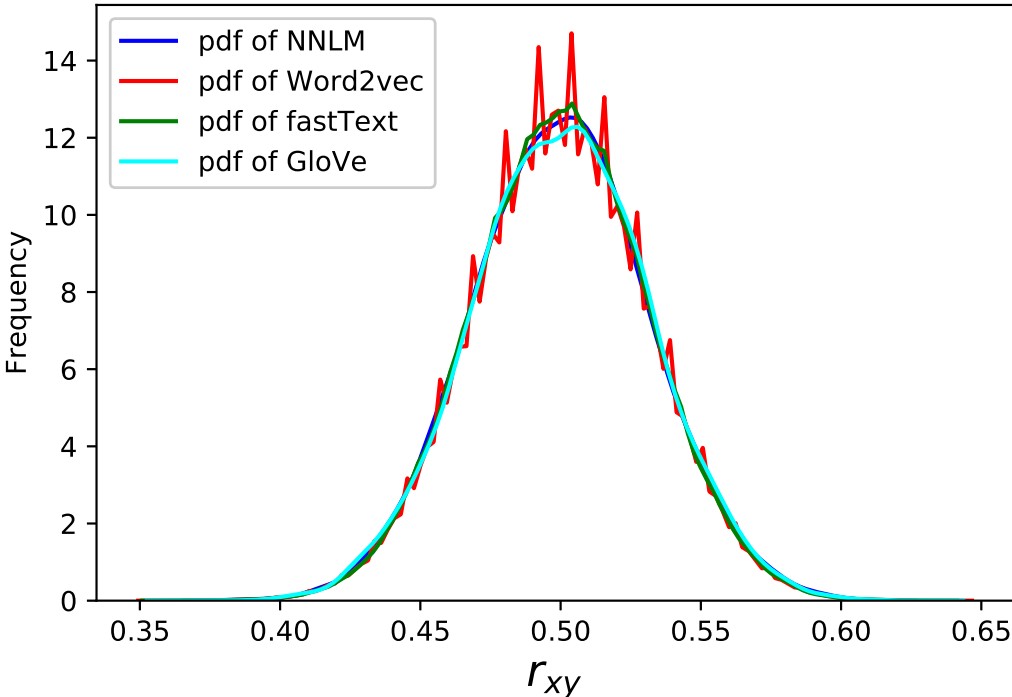

**Figure 19.** Generality about models to be applied.

## 11. Limitations

Though novel it is, TEDL suffers from some drawbacks. Firstly, security has not been theoretically proved, since it is hard to interpret the variation of parameters in deep learning model. Secondly, the choices on the kind of initial address are limited, for the sake of key sensitivity. Thirdly, the efficiency of encryption and decryption is negatively correlated to the number of entries in the codebook, for the process mainly involves lookup operations. In addition, in spite of almost impossible, two different words may map to the same hash, which means the results of inverted indexing may be more than one. Under this case, the decryption should depend on the context to select the correct plaintext. Besides, the first stage takes too long, during which the communication must suspend, bringing inconvenience. Moreover, some requirements to models should be met, which is detailed in Section 6.2. Especially, models must completely eliminate randomization. Finally, the self-updating mechanism in this paper remains to be improved.

## 12. Conclusions and Future Work

In this paper, we propose a new text encryption method based on deep learning named TEDL. It is the first time to directly apply deep learning model to encryption, mainly utilizing the uninterpretability and time-consuming training features. The time-varying and self-updating characteristics of TEDL deal with the problem of key redistribution and the two-stage structure makes it hard to carry out brute-force attack and makes it efficient for communication. Moreover, TEDL bears other superior properties such as anti-interference, diffusion and confusion, high sensitivity, generality and so on, all of which have been confirmed through experiments.

It is worth mentioning that both encrypted objects and models are expandable. Objects in various forms, such as binary numbers, texts, images, videos, or even multimodal information, can be encrypted with TEDL. For example, assuming TEDL adopts word embedding model and aims to encrypt binary numbers, a binary library can be constructed serving as public corpus while additional text in either original or binary form acts as an incremental corpus, then training performs on the synthetic corpus. Inspired by the fact that more and more objects can be embedded (e.g., the network [55]), it is natural that those

objects can be encrypted by TEDL with embedding model, for which we might as well name as embedding encryption.

As for the extensibility of models, all the models that satisfy the Model Requirements in Section 6.2 can be employed by TEDL, and it is easy to meet those requirements. For example, nearly all deep learning models own public training set and it is easy to get whether texts, images or videos on the Internet as long as their corresponding addresses exist, meeting Model Requirements 1 and 2. Besides, considerable models, such as CNN and LSTM, own a large number of parameters, satisfying Model Requirement 3. Moreover, as for supervisory models, they are still available, as long as labels are preconcerted without the necessity of secret, thus satisfying Model Requirement 4. Finally but not least, TEDL cannot only use for encryption. From the perspective of generating a key stream, such a large number of parameters in the deep learning model may be utilized, which is also worth exploring.

**Author Contributions:** Conceptualization, P.W. and X.L.; methodology, P.W. and X.L.; validation, X.L.; writing–original draft preparation, X.L.; writing–review and editing, P.W. All authors have read and agreed to the published version of the manuscript.

**Funding:** The work is supported by National Key R&D Program of China (2018YFD1100302), National Natural Science Foundation of China (No.61972082, No.72001213), and All-Army Common Information System Equipment Pre-Research Project (No.31511110310, No.31514020501, No.31514020503).

**Institutional Review Board Statement:** Not applicable.

**Informed Consent Statement:** Not applicable.

**Data Availability Statement:** Data sharing not applicable.

**Conflicts of Interest:** The authors declare no conflict of interest.

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
