# Peer review of "TEDL: A Text Encryption Method Based on Deep Learning"

_applsci, doi:10.3390/app11041781_

Round 1
Reviewer 1 Report
This is a fascinating paper that implements Word2Vec into a cryptographic model. As I understood it, the codebook is self-organizing, rather than being defined in advance. From my own understanding, this is a significant and important turn in communicational paradigms that are impacting several disciplines. The article is very technical, but the level of detail is necessary for anyone who would try to test the model proposed by the authors for themselves. The link to the Github library of the authors is a precious addition for anyone wanting the examen the python code in detail and run it for themselves.
Because some readers may want to skip the definitions and equations and jump to the overall pipeline, the figures summarizing the certain aspects of the model could be improved. Figure 1 works well, but Figures 2 and 3 need a common legend, the U and the v (Greek upsilon?) should correspond. It is not clear how a corpus graph can be contained within an incremental corpus graph as the two concentric circles suggest. Figure 3 is illustrating a 'construction' and therefore is in time. What is the time vector? From inside to outside? This is not clear and reads more to me as something in space than in time.
The paper would be even stronger if there was room for a demonstration of a communicational setup in which two parties are trying to exchange coded messages. The demonstration given in the paper is only for the performance of the cryptographic model, not a real-world situation of its 'application'.
Otherwise, this paper appears to me to be ready to be shared with the scientific community.
Author Response
*We thank the anonymous reviewers for their careful reading of our manuscript and their many insightful comments and suggestions, which is very important for improving our manuscript. Below we respond to the comments of each reviewer in detail.
Response to comments of Reviewer 1
- Questions about the figures in the model.
- Because some readers may want to skip the definitions and equations and jump to the overall pipeline, the figures summarizing the certain aspects of the model could be improved.
- Figure 1 works well, but Figures 2 and 3 need a common legend, the U and the v (Greek upsilon?) should correspond.
- It is not clear how a corpus graph can be contained within an incremental corpus graph as the two concentric circles suggest.
- Figure 3 is illustrating a 'construction' and therefore is in time. What is the time vector? From inside to outside? This is not clear and reads more to me as something in space than in time.
Response:
Thank you for your comments and suggestions.
As you pointed out, because there are some theoretical definitions and formulas in this paper, it would take some time for readers to fully understand our model. Therefore, we very much agree that the figures are very important for readers to understand how our model works. Clear and simple figures could help readers better understand our paper. In the revision version, we try our best to improve the figures about model.
In the revision version, we have improved Figure 2 and Figure 3. First, the common legend is used. Second, in order to unify symbols, we use $V_j^i$ to denote the j-th unit (paper) of the i-th layer in incremental corpus. Moreover, in order to make the figures express our ideas more clearly, we use three different colors to distinguish three different corpuses: synthetic corpus, original corpus, and incremental corpus. And we also use two colors to distinguish two graphs: initial incremental corpus graph, and incremental corpus graph. A graph is an abstract concept. A corpus is an entity.
The incremental corpus graph corresponds to the incremental corpus. And the incremental corpus graph is extended from the initial incremental corpus graph. In other words, as Figure 3 shows, a corpus graph can be contained within an incremental corpus.
For the Figure 3, as you said, Figure 3 is illustrating a construction process. Although it is in time, it belongs to stage of pre-process, so the time is not the matter the model should concern. And in term of the time, what you said is right. It is something in space, not in time.
By the way, readers could concern the setup time and codebook updating time in our model. The setup time is corresponding to Figure 10. But it is not so important in applications, the main reason is that the setup stage can be viewed as a pre-processing stage. And it only needs to be done once during a relatively long time, in other words, we do not invoke setup as frequent as encryption and decryption. For the updating time, the codebook updating stage is similar to setup stage in essential. Training dominates the updating time. What’s more, the self-updating stage illustrated in section 6 only needs to be done once during a relative long period. The self-updating is actually another setup stage. As for the security of codebook, it is also guaranteed by the time-varying characteristic, which enforces another kind of “update”. And the time corresponding to this part is included in the performance shown in Figure 18.
- Questions about the demonstration.
- The paper would be even stronger if there was room for a demonstration of a communicational setup in which two parties are trying to exchange coded messages. The demonstration given in the paper is only for the performance of the cryptographic model, not a real-world situation of its 'application'.
Response:
Thank you for your comments and suggestions.
In order to illustrate the principle of our model, we have carefully designed a general demonstration or example throughout the paper. The demonstration is not only used to analyze the performance of the cryptographic model, but also the detail principle of the proposed model. In the demonstration, as the two parties communicates, the sender will find the encrypted form of messages in codebook word by word, and replace the encrypted word with the original word to send to the receiver. And the receiver will find the decrypted form of messages word by word, decrypting the ciphertext to plaintext. During the communication, the codebook varies in a certain manner, to avoid same plaintexts mapping to same ciphertexts.
We agree that the current demonstration in this paper is not from real-world application, but it is representative enough for most application scenarios that we can image.
We are looking for some real applications to verify our model. But it would need some time. Meanwhile, we will provide more real-world application examples in the GitHub project to show the encryption performance of our model.

Reviewer 2 Report
The paper is well written and well organized. The subject of the paper is very important. In case that there are similar works, some comparison in term of safety will improve the quality of the paper.
In addition, some more comments about the sha256 selection (eg instead of some newer and safer) should be mandatory.
Author Response
*We thank the anonymous reviewers for their careful reading of our manuscript and their many insightful comments and suggestions, which is very important for improving our manuscript. Below we respond to the comments of each reviewer in detail.
Response to comments of Reviewer 2
- Questions about the safety comparison with similar or related works.
- The subject of the paper is very important. In case that there are similar works, some comparison in term of safety will improve the quality of the paper.
Response:
Thank you for your suggestions and comments.
The safety of TEDL is based on the fact that the deep learning model is hard to explain. And it hard to completely prove its safety in theory, as well as to do safety comparison. However, as the interpretability of learning model is regarded as difficult task in research, we are confident about its safety now and we keep trying to prove it.
In the last paragraph of Section 2, we analyze the relationship between safety and current interpretability research on deep learning models. Moreover, in Section 9, we analyze the security comparison with some related works during the encryption and decryption process.
Besides the safety, for the efficiency comparison, it can be seen that for the same amount of plaintext, TEDL needs least time, which means that TEDL is fast, in other words, it shows that our method is efficient.
- Questions about comments on the sha256 selection.
- Some more comments about the sha256 selection (eg instead of some newer and safer) should be mandatory.
Response:
Thank you for your suggestions.
Besides the safety, performance and space expansion are the other two aspects we need to consider. When encrypting, the codebook is changing in real time. Therefore, as keeping the proper safety, we need to choose quicker algorithm. For example, as the following table shows, which means that SHA-256 is faster than those in SHA3.
Meanwhile, although there are some algorithms which are safer and faster than SHA-256, the ciphertext will occupy more space. For example, SHA-512 will occupy space as two times as SHA-256. In Section 6.4, we give the comments on the SHA-256 selection based on the above analysis.
